# Ribosomal protein L5 facilitates rDNA-bundled condensate and nucleolar assembly

Haruka Matsumori[1,*], Kenji Watanabe[2,*], Hiroaki Tachiwana[2], Tomoko Fujita[2], Yuma Ito[3], Makio Tokunaga[3], Kumiko Sakata-Sogawa[3], Hiroko Osakada[4], Tokuko Haraguchi[4,5], Akinori Awazu[6,7], Hiroshi Ochiai[6], Yuka Sakata[2], Koji Ochiai[8], Tsutomu Toki[9], Etsuro Ito[9], Ilya G Goldberg[10], Kazuaki Tokunaga[1], Mitsuyoshi Nakao[1], Noriko Saitoh[2]

The nucleolus is the site of ribosome assembly and formed through liquid–liquid phase separation. Multiple ribosomal DNA (rDNA) arrays are bundled in the nucleolus, but the underlying mechanism and significance are unknown. In the present study, we performed high-content screening followed by image profiling with the wndchrm machine learning algorithm. We revealed that cells lacking a specific 60S ribosomal protein set exhibited common nucleolar disintegration. The depletion of RPL5 (also known as uL18), the liquid–liquid phase separation facilitator, was most effective, and resulted in an enlarged and un-separated sub-nucleolar compartment. Single-molecule tracking analysis revealed less-constrained mobility of its components. rDNA arrays were also unbundled. These results were recapitulated by a coarse-grained molecular dynamics model. Transcription and processing of ribosomal RNA were repressed in these aberrant nucleoli. Consistently, the nucleoli were disordered in peripheral blood cells from a Diamond–Blackfan anemia patient harboring a heterozygous, large deletion in *RPL5*. Our combinatorial analyses newly define the role of RPL5 in rDNA array bundling and the biophysical properties of the nucleolus, which may contribute to the etiology of ribosomopathy.

## Introduction

The nucleolus is a membrane-less nuclear organelle for ribosome biogenesis, where ribosomal DNA (rDNA) transcription and ribosome particle assembly occur (Turowski & Tollervey, 2015). The nucleolus contributes to protein translation that is directly linked to cell proliferation, and is responsive to various environmental changes and diseases (Shav-Tal et al, 2005; Mekhail et al, 2006; Boulon et al, 2010; Tanaka et al, 2010; Audas et al, 2012). Under stress conditions, the nucleolus loses its integrity, in a process termed "nucleolar stress" (Shav-Tal et al, 2005; Boulon et al, 2010; Audas et al, 2012). Altered nucleolar shapes and sizes have also been observed in several diseases, including cancers and neurodegenerative disorders, for which they serve as good diagnostic indicators (Derenzini et al, 2009). However, the mechanism of nucleolar formation is largely unclear (Misteli, 2001). It remains elusive whether the nucleolus is a simple by-product of massive transcription and protein assemblies in ribosome biogenesis, or it plays an active regulatory role in orchestrating ribosome synthesis.

In eukaryotes, the nucleolus has a tripartite structure composed of the fibrillar center (FC), the dense fibrillar component (DFC), and the peripheral granular component (GC). rDNA is transcribed at the border of the FC and DFC by RNA polymerase I (Pol I) with its transcription factors UBF and TIF-IA. The early steps of precursor rRNA (pre-rRNA) processing take place in the DFC, and later steps occur in the GC to produce three mature rRNAs (18S, 5.8S, and 28S) that are then complexed with ribosomal proteins (RPs) to form the two ribosomal subunits, 40S and 60S (McStay & Grummt, 2008; Drygin et al, 2010). The 40S small subunit contains the 18S rRNA and 33 ribosomal proteins (RPSs), whereas the 60S large subunit contains the 5.8S, 5S, and 28S rRNAs, and 47 ribosomal proteins (RPLs). Once the ribosome subunits are matured, they are exported to the cytoplasm where they engage in translation (Turowski & Tollervey, 2015). Ribosome dysfunction is associated with various diseases, in a family collectively called "ribosomopathies." Its founding member is Diamond–Blackfan anemia (DBA), a congenital bone marrow failure syndrome characterized by anemia, developmental abnormalities and insufficient production of erythroid

[1]Department of Medical Cell Biology, Institute of Molecular Embryology and Genetics, Kumamoto University, Kumamoto, Japan  [2]Cancer Institute of Japanese Foundation for Cancer Research, Tokyo, Japan  [3]School of Life Science and Technology, Tokyo Institute of Technology, Yokohama, Japan  [4]Advanced ICT Research Institute Kobe, National Institute of Information and Communications Technology, Kobe, Japan  [5]Graduate School of Frontier Biosciences, Osaka University, Osaka, Japan  [6]Graduate School of Integrated Sciences for Life, Hiroshima University, Higashi-Hiroshima, Japan  [7]Research Center for the Mathematics on Chromatin Live Dynamics (RcMcD), Hiroshima University, Higashi-Hiroshima, Japan  [8]DAIZ Inc, Kumamoto, Japan  [9]Department of Pediatrics, Hirosaki University Graduate School of Medicine, Hirosaki, Japan  [10]Image Informatics and Computational Biology Unit, Laboratory of Genetics, National Institute on Aging, National Institutes of Health, Baltimore, MD, USA

Correspondence: noriko.saito@jfcr.or.jp; mnakao@gpo.kumamoto-u.ac.jp
*Haruka Matsumori and Kenji Watanabe are co-first author.

precursors. The molecular mechanisms underlying these diseases are largely unknown (McGowan & Mason, 2011; Kuramitsu et al, 2012; Farrar et al, 2014; Wang et al, 2015); however, only a subset of RPs is responsible for the diseases, suggesting the differential roles among RPs.

The nucleolus forms around arrays of rDNA sequences, termed nucleolar organizer regions (NORs). The nucleolus represents the largest site of transcription in the nucleus, as rDNA transcription accounts for 35–60% of total transcription in actively cycling eukaryotic cells (Moss and Stefanovsky [2002] and references therein). In human, NORs are located on chromosomes 13, 14, 15, 21, and 22 of two alleles from both parents, accounting for 10 chromosome sites (McStay & Grummt, 2008; Németh et al, 2010; van Koningsbruggen et al, 2010). They are distributed to 3–4 nucleoli in a cell, implying that multiple NORs are bundled in each nucleolus. This spatially dense positioning of the NORs may contribute to their massive transcription and processing capacities.

Studies in *Xenopus* oocytes revealed that the nucleolus is spontaneously formed through liquid–liquid phase separation (LLPS), a biophysical process driven by intrinsically disordered regions or low complexity sequences within their molecular components (Feric & Brangwynne, 2013; Mitrea et al, 2016). The nucleolus represents a biophysical state called a liquid droplet with high viscosity, where RNA and proteins are highly condensed while allowing dynamic molecular interactions (Feric & Brangwynne, 2013; Zhu et al, 2019). Other in vitro studies suggested that nucleophosmin (NPM1), an abundant oligomeric protein in the GC, facilitates nucleolar assembly through phase separation by multivalent interactions with itself and with RPs and rRNAs (Mitrea et al, 2014, 2016; Nicolas et al, 2016). The question of whether the mechanisms suggested by these studies in vitro, as well as in *Xenopus laevis* oocytes, are shared with somatic nucleoli remains to be answered, as there are far fewer somatic nucleoli, and they are much smaller and tethered to chromosomes.

The structure of the nucleolus is amorphous and highly variable, even within the same cell type. Conventional strategies for image analysis involve the predefinition of specific objects to automatically recognize and measure their morphological features. An excellent image analysis system, the iNo scoring method, which is specialized for analyses of the nucleolar morphology, was recently developed and offers unprecedented statistical power to identify factors important for nucleolar structure maintenance (Nicolas et al, 2016; Stamatopoulou et al, 2018). In this study, we used a supervised machine learning algorithm, "wndchrm" (weighted neighbor distances using a compound hierarchy of algorithms representing morphology) to profile morphologies. It is distinct from the iNo scoring method because it is not limited to analyses of the nucleolus but can be used for any structures, since it was developed for population-based image classification, similarity measurements and other purposes (Shamir et al, 2008a). Instead of specifying target morphologies and choosing particular algorithms, wndchrm users define classes by providing multiple image examples for each class, such as knockdown and control cells, for example. After machine learning, the wndchrm program computes the degrees of morphological similarity as a distance in the feature space resulting from multiple rounds of classification by cross validation tests. It has been successfully used to investigate many problems (Shamir et al, 2008a, 2008b), including reprogramming of human iPS colonies (Tokunaga et al, 2014), early detection of osteoarthritis (Shamir et al, 2009a, 2009b), measurement of sarcopenia in *Caenorhabditis elegans* (Johnston et al, 2008), and classification of malignant lymphoma (Shamir et al, 2008b).

In this study, we performed high-content siRNA screening combined with wndchrm, to identify the factors required for NPM1 morphology in the nucleolus. The results revealed that the depletions of the selected RPs share common changes, and the RPL5 depletion exerted the greatest effect. Our multidisciplinary studies further clarified the detailed roles of RPL5 in the nucleolus, including the bundling of the rDNA array, the biophysical properties and the rRNA production and processing. We then explored the nucleolar morphology in DBA patients with mutations in RPL5. Our study demonstrates a novel function of RPL5 in the higher order structure of the rDNA array and suggests the pathological basis of red cell aplasia.

# Results

### The contributions of specific 60S ribosome proteins to the nucleolar morphology revealed by high-content siRNA screening

To identify the cellular proteins that are involved in the nucleolar structure, we performed high-content siRNA screening with an siRNA library that targeted 745 human genes (Table S1). The targeted genes are involved in nuclear events and signaling pathways for gene regulation, energy metabolism and DNA repair (Boulon et al, 2010), together with nucleolar components that had been previously identified by a proteome analysis (Andersen et al, 2005). We introduced each siRNA into HeLa cells for 48 h, and then visualized the nucleoli by immunofluorescence with antibodies against NPM1, a component of the GC region of the nucleolus (Fig S1A). To secure each knockdown, we used a method with pooled siRNAs including four unique siRNAs designed for one target gene, which guarantees a targeted knockdown. To monitor the appropriate RNAi knockdowns, each set of experiments always contained a pair of negative (siGL3, non-targeting sequence) and positive (si*LMNA*) controls (see the Materials and Methods section for details). For each knockdown, images containing ~100 cells were captured in triplicate experiments, and the intensities and areas of the NPM1 signals were subjected to quantification (Fig S1A). As a blind control of this screen, the library included the siRNA for NPM1, which diminished NPM1 signals as expected (Figs 1A and S1B). siRNA against TIF-IA, a transcription initiation factor of RNA Pol I, also reduced NPM1 (Figs 1A and S1B). This was consistent with previous reports showing that the inhibition of RNA Pol I with actinomycin D disrupted nucleolar formation (Shav-Tal et al, 2005; Boulon et al, 2010). Accordingly, these results validated our high-content siRNA screening experiments.

Among the examined factors, we found that the individual loses of 16 proteins reproducibly changed the nucleolar intensities and shapes, based on visual-inspections. They included RPs, a gene repressor, mRNA splicing factors, transcription factors, and signal transducers (Fig 1A and Table 1). Knockdowns of representative

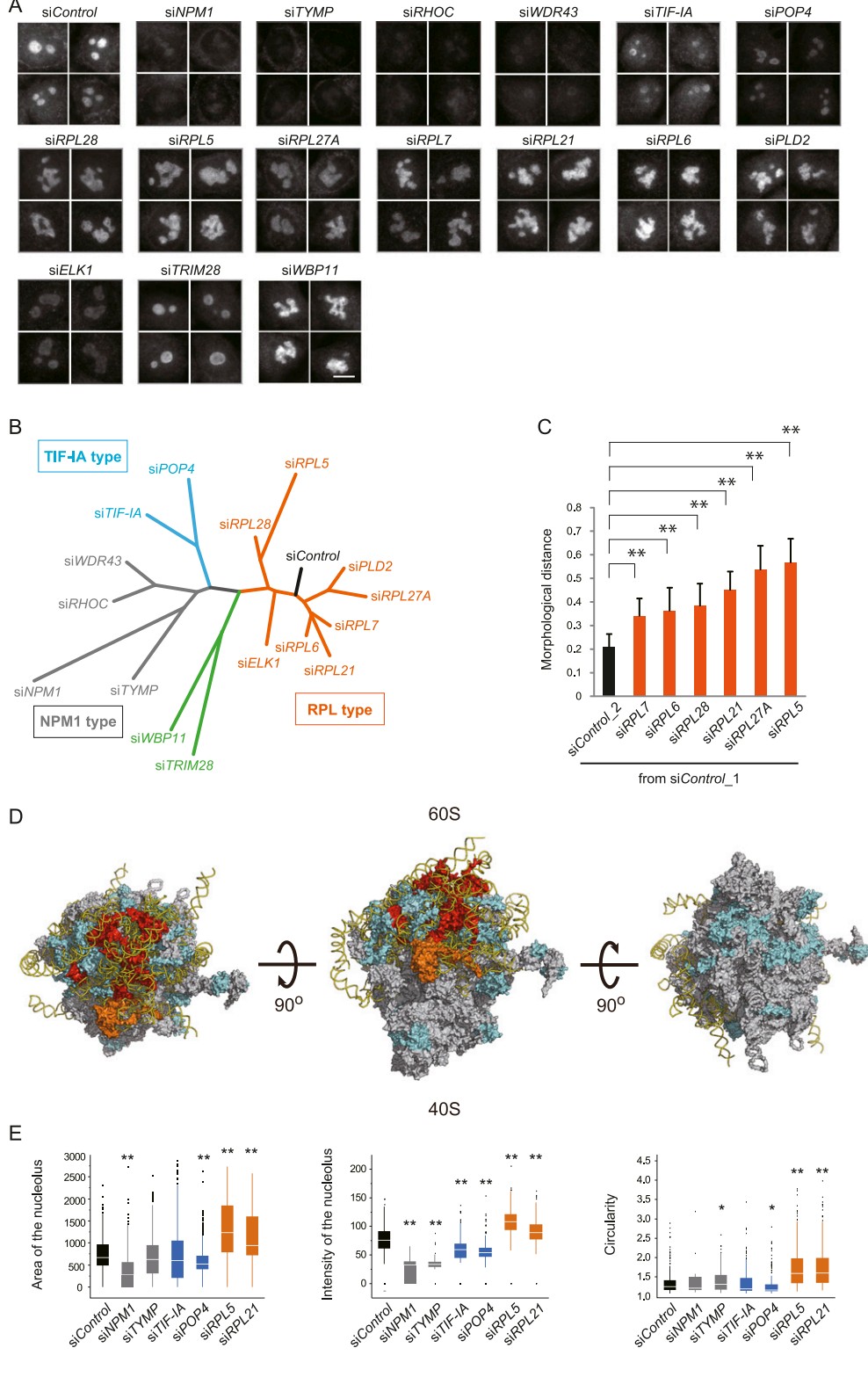

**Figure 1. Morphological profiling reveals a nucleolar role for 60S ribosomal proteins.**
**(A)** Immunofluorescence analyses of HeLa cell nucleoli after knockdowns of the 16 indicated genes. HeLa cells were transfected with specific siRNAs, and nucleoli were visualized using anti-nucleophosmin (NPM1) antibodies. Scale bar, 10 μm. **(B)** Dendrogram showing the similarity of the nucleolar morphologies among the 16 knockdowns. NPM1 image sets were analyzed with the wndchrm program (each n = 80 images, Fig S2). The 16 knockdowns were classified into groups: NPM1, transcription initiation factor (TIF-IA), 60S ribosomal protein (RPL) types, and others. **(C)** The morphological distances (MDs) from the control (siControl_1). A set of control images was divided into two subsets, siControl_1 and _2, which were used as a reference and a negative control, respectively (n = 40 images). Values are the means ± SD of 20 cross validation tests. **P < 0.01. **(D)** Three-dimensional models of a human ribosome particle composed of the large (60S) and small (40S) subunits are shown. The structure in the middle is virtually tilted in both directions at 90 degrees (left and right). RPL5 is color-coded in orange, and the RPLs whose knockdowns resulted in aberrant nucleoli are colored red. RPs that were included in the siRNA screening, but did not show an unusual nucleolar phenotype are colored blue. Other RPs are grey. **(E)** Quantifications of the nucleoli visualized with the NPM1 antibodies. Area, signal intensity and circularity were measured (n > 100 cells). Loss of RPL5 and RPL21 resulted in a nucleolar size increase and a sphericity (circularity) loss. The boxes represent the median, 25%, and 75% percentiles, and black dots are outliers. *P < 0.05; **P < 0.01.

genes were confirmed by qRT-PCR (Fig S1C). We also verified that characteristic nucleolar morphologies appeared using single siRNAs against RPL5 (siRPL5 #1 and siRPL5 #2) and RPL21 (siRPL21 #1 and siRPL21 #2) (Fig S1D).

To quantitatively evaluate the similarities and differences among the nucleolar morphologies exhibited by the knockdowns of the 16 genes, we collected 80 NPM1 immunofluorescence images for each knockdown and analyzed them with the wndchrm program

**Table 1. A list of factors whose knockdowns resulted in aberrant nucleoli**

| Gene name | Protein name | GO | Localization | UniProt Acc. |
|---|---|---|---|---|
| NPM1 | Nucleophosmin | Chaperon, host-virus interaction, RNA-binding | Nucleus/nucleolus cytoplasm | P06748 |
| TYMP | Thymidine phosphorylase | Developmental protein, glycosyltransferase, growth factor, angiogenesis | Cytosol | P19971 |
| RHOC | Rho-related GTP-binding protein RhoC | GTP binding, signal transducer activity | Cell membrane | P08134 |
| WDR43 | WD repeat-containing protein43 | poly(A) RNA binding | Nucleus/nucleolus | Q15061 |
| TIF-IA | RNA polymerase I-specific transcription initiation factor | Transcription regulation | Nucleus/nucleolus | Q9NYV6 |
| POP4 | Ribonuclease P protein subunit p29 | Hydrolase, tRNA processing | Nucleus/nucleolus | O95707 |
| RPL5 | 60S ribosomal protein L5 | Ribonucleoprotein, ribosomal protein | Cytoplasm nucleus/nucleolus | P46777 |
| RPL6 | 60S ribosomal protein L6 | Ribonucleoprotein, ribosomal protein | Cytoplasm, nucleus | Q02878 |
| RPL7 | 60S ribosomal protein L7 | Ribonucleoprotein, ribosomal protein | Cytoplasm, nucleus | P18124 |
| RPL21 | 60S ribosomal protein L21 | Ribonucleoprotein, ribosomal protein | Cytoplasm nucleus/nucleolus | P46778 |
| RPL27A | 60S ribosomal protein L27a | Ribonucleoprotein, ribosomal protein | Cytosol | P46776 |
| RPL28 | 60S ribosomal protein L28 | Ribonucleoprotein, ribosomal protein | Cytoplasm | P46779 |
| PLD2 | Phospholipase D2 | Hydrolase, lipid degradation, lipid metabolism | Cell membrane | O14939 |
| ELK1 | ETS domain-containing protein Elk-1 | Activator, transcription, transcription regulation | Nucleus | P19419 |
| TRIM28 | Transcription intermediary factor 1-beta | Ligase, repressor, transcription regulation, Ubl conjugation pathway | Nucleus | Q13263 |
| WBP11 | WW domain-binding protein 11 | mRNA processing, mRNA splicing, rRNA processing | Cytoplasm, nucleus | Q9Y2W2 |

(Fig S2). For the machine learning, 2,873 feature values were computed for every image, followed by automatic extraction of the relevant features that can discriminate classes (Table S2). By multiple cross validation tests, class probability matrices were produced to visualize the morphological relations as a dendrogram (Fig 1B and C, also see the Materials and Methods section).

The resulting dendrogram was branched into four groups: the NPM1-type including siNPM1, siTYMP, siRHOC, and siWDR43, the TIF-IA-type including siTIF-IA and siPOP4, the RPL-type including ribosomal proteins of the large subunit (siRPLs), and others including siTRIM28 and siWBP11 (Fig 1B). Only some of the RP knockdowns showed nucleolar changes (Fig S3A). Among the 34 RPs included in our siRNA library, six knockdowns showed morphological changes (Fig 1A–C), and most of them function in late steps of rRNA processing (Robledo et al, 2008; Gamalinda et al, 2014). Other RPs involved in different steps had minimal effects, even though their mRNA expression was reduced to a similar extent as the RPL5 mRNA by the siRNA treatments (RPL9, RPL13 and RPS9 in Fig S3B). Our analysis of the three-dimensional models based on the Protein Data Bank (PDB ID: 4V6X) showed that the influential RPs are located near each other on the 60S large subunit of the ribosome particle (Fig 1D).

With windchrm, we also measured the pair-wise morphological distance (MD) values from the control (siControl), indicating the degree of nucleolar shape. We found that the changes in each RPL knockdown were significant, and the one in RPL5 knockdown cells was the largest (Fig 1C). We also measured nucleolar morphology parameters, including area, intensity, and circularity, which showed that the RPL knockdowns (siRPL5 and siRPL21) resulted in enlarged, non-spherical nucleoli (Fig 1E).

Usually, cells have several nucleoli per nucleus, but most of the RPL knockdowns had an un-separated large nucleolus. These results suggest that the biophysical features of a liquid droplet may be significantly lost in RPL knockdown cells.

Because the morphological changes were most significant in the RPL5 knockdown cells (Fig 1B, C, and E), we investigated RPL5 in more detail. We were interested in how the p53 tumor suppressor is involved in the RPL5 phenotype. Our immunoblot analyses showed that in HeLa cells, the p53 level was decreased with the RPL5 knockdown (Fig S3C, left). This is consistent with the previous report that RPL5 blocks MDM2-mediated p53 ubiquitination and degradation upon nucleolar stress (Dai & Lu, 2004). Because HeLa cells have an impaired p53 signaling pathway because of E6 viral proteins, we also analyzed RPE-1 cells. These cells are karyotypically normal and have a functional p53 signaling pathway. In fact, in RPE-1 cells, upon the depletion of TIF-IA due to nucleolar stress, strong p53 protein accumulation was accompanied with p21 accumulation (Fig S3C, middle, Fig S3C, right). In these cells, the p53 protein was not remarkably accumulated with the RPL5 depletion, even though, we observed the aberrant nucleolar morphologies in the RPL5-depleted cells (Fig S3D). Cell growth was inhibited with the RPL5 knockdown in both the HeLa and RPE-1 cell lines (Fig S3E and F).

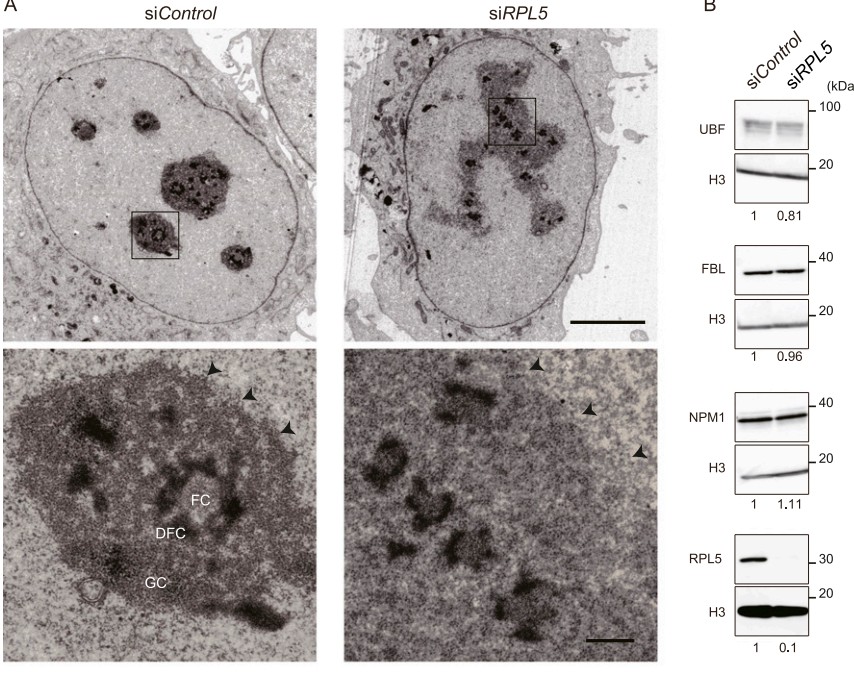

A  siControl  siRPL5

B

C

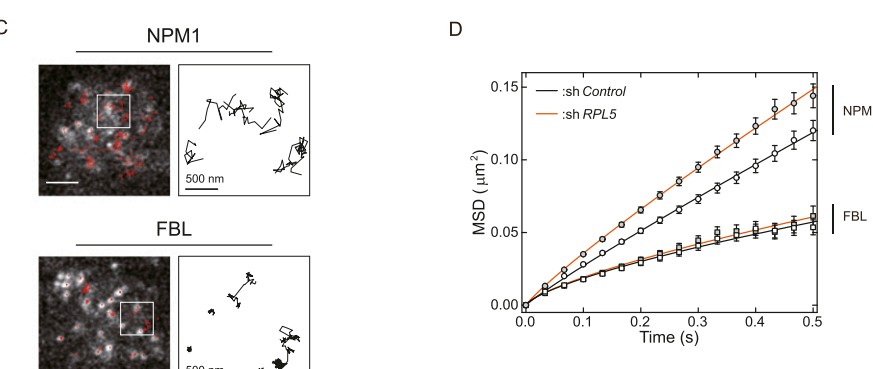

NPM1

500 nm

FBL

500 nm

D

MSD ($\mu m^2$) vs Time (s)

— :sh Control
— :sh RPL5

NPM1

FBL

**Figure 2. Specific disintegration and mobility loss of the GC region after the RPL5 knockdown.**
**(A)** Electron microscopy images of control and RPL5-depleted cells. The square portions in the upper panels are magnified in the lower panels. In the GC of the RPL5-depleted cell, the sphericity is lost, the granular density is decreased, and the boundary is less clear (triangles). Scale bars, 5 $\mu$m (top) and 0.5 $\mu$m (bottom). **(B)** Immunoblot analyses of the UBF, FBL, NPM1, and RPL5 proteins in the indicated cells. Band intensities relative to siControl are shown below. Histone H3 was used as an internal control. **(C)** Single-molecule trajectory analyses of NPM1 and FBL. SNAP-tagged NPM1 and FBL expressing HeLa cells were labeled with 3 nM MR-Star substrate and observed under HILO fluorescence microscopy. The trajectories of single molecules (red lines) are superimposed on the snapshots from the movies on the left. Scale bar, 2 $\mu$m. Representative trajectories are enclosed within white boxes and enlarged on the right. Scale bar, 500 nm. **(D)** Mean squared displacement values of NPM1 and FBL along the time for single-molecule trajectories in the indicated cells (RPL5 shRNA knockdown and shRNA control). Values are means ± SEM from more than 790 trajectories in 20 cells. The loss of RPL5 accelerated the mobility of NPM1 in GC.

These results suggest that the cell growth inhibition and aberrant nucleolar morphology upon the RPL5 knockdown are independent of the p53 signaling pathway.

We then asked whether the nucleolar changes in the RPL-knockdown cells are simply due to dysfunctional protein translation. We treated cells with cycloheximide (CHX), a protein synthesis inhibitor, at 50 $\mu$g/ml for 6 h (Nolop & Ryan, 1990; Haaf & Ward, 1996; Schneider-Poetsch et al, 2010), but did not detect significant nucleolar alterations (Fig S3G). Similarity measurements with wndchrm showed that the CHX treatment barely changed the nucleolus, as also observed in treatments with DMSO, siControl or both (Fig S3H and Table S3).

Our live cell imaging demonstrated that the aberrant nucleoli appear after cells exit mitosis (Fig S4A and B), when the nucleoli are reconstructed from disintegrated and dispersed components (Hernandez-Verdun, 2011). Therefore, RPL5 may be required for the initiation of nucleolar formation. Finally, the RPL5 knockdown did not affect other nuclear bodies, such as Cajal bodies and nuclear speckles, based on visual-inspections (Fig S4C). Altogether, these

results indicated that RPL5 plays an essential role in nucleolar formation.

## RPL5 contributes to the structural and biophysical properties of the GC region of the nucleolus

To investigate whether the overall nucleolar integrity is affected by the depletion of RPL5, we observed RPL5-depleted cells by transmission electron microscopy (Fig 2A). The control cells showed the characteristic tripartite organization, in which the FC was surrounded by an electron-dense DFC, which resided in an elliptically distributed GC. In RPL5 knockdown cells, the FC and DFC retained their structural integrity; however, they were smaller, fragmented, and scattered throughout the disorganized GCs, which were spread over the nucleoplasm to make individuals to fuse with each other. The surface of the GC was convex and concave, and the granular density was decreased inside the GC, and the boundary between the GC surface and the nucleoplasm was less distinct. This may be due to the decreased density of mature ribosomal subunits,

which appear as small dark granules on transmission electron microscopy labeled with uranyl acetate. Our immunoblot analyses showed that the RPL5 knockdown in this time window (48 h) did not affect protein expression in the nucleolus, probably due to the excess amount of preexisting translational machineries (Fig 2B). These findings demonstrated that the nucleolar structural change is due to the re-distribution of the nucleolar components.

The lower molecular density, the loss of sphericity, and the less distinct interface of the GC region suggested that its biophysical nature might be altered in the absence of RPL5, as shown in vitro (Mitrea et al, 2014, 2016). We therefore investigated the molecular dynamics of the nucleolus in living HeLa cells, by single-molecule trajectory analyses (Feric & Brangwynne, 2013; Ito et al, 2017). We expressed SNAP-tagged NPM1 and FBL in HeLa cells to monitor GC and DFC, respectively. We fluorescently labeled them with a tetramethylrhodamine derivative (TMR-Star), for observations by HILO fluorescence microscopy (Fig 2C and Videos S1–S4) (Tokunaga et al, 2008). For each, more than 2,000 trajectories in 60 cells were measured to quantify dynamics, and the mean squared displacement (MSD) values were calculated (Fig 2C and D and Table S4). The diffusion coefficient $D$ of NPM1, 0.054 ± 0.002 $\mu m^2/s^\alpha$, mean ± SD, the same below, was significantly larger than that of FBL, 0.023 ± 0.004 $\mu m^2/s^\alpha$, in control cells (sh$Control$), consistent with a previous in vitro study (Feric et al, 2016). The diffusive exponent $\alpha$ of NPM1 (0.88 ± 0.04 in living cells, this study, and 0.92 ± 0.06 in vitro [Feric et al, 2016]) was closer to 1.0, as compared with that of FBL (0.69 ± 0.09 in living cells and 0.50 ± 0.10 in vitro [Feric et al, 2016]). These results demonstrated that the biophysical nature was shared between the somatic cellular nucleolus and the in vitro–reconstituted liquid droplets (Feric et al, 2016).

The previous in vitro study (Feric et al, 2016) also estimated that the viscosity $\eta$ of the GC was 0.74 ± 0.06 Pa s, using the diffusion coefficient $D$ and Stokes–Einstein equation. The diffusion coefficient of NPM1 (0.054 $\mu m^2/s^\alpha$) gave a $\eta$ of 0.7 Pa·s when the radius is presumed to be 6 nm, which is larger than the radius of 3 nm of the NPM1 pentameric ring (Lee et al, 2007). This strongly suggested that the NPM1 diffuses involving intermolecular interactions. Therefore, we speculated that RPL5 is one of the contributors because it interacts with NPM1 (Yu et al, 2006; Mitrea et al, 2016). In fact, in the RPL5 knockdown cells, the mobility of NPM1 in the GC was accelerated: RPL5 shRNA knockdown, $D$ = 0.067 ± 0.001 $\mu m^2/s^\alpha$; shRNA control, $D$ = 0.054 ± 0.002 $\mu m^2/s^\alpha$, mean ± SD, respectively (Fig 2D and Videos S3 and S1). In contrast, this knockdown had little effect on the mobility of the FBL protein in the DFC. Therefore, RPL5 represses the NPM1 mobility to confer anomalous diffusion in control cells. Single-molecule tracking analyses demonstrated a novel contribution of RPL5 to a biophysical property of the GC region in the nucleolus of the somatic cell.

## RPL5 bundles the rDNA arrays in the nucleolus

In mammalian somatic cells, multiple rDNA arrays are bundled and tethered to the nucleolus (McStay & Grummt, 2008; Németh et al, 2010; van Koningsbruggen et al, 2010). This reflects how the nucleolus is re-formed at the exit from mitosis. At the late stage of mitosis, Pol I transcription resumes and nucleoli begin to reform

around individual rDNA arrays, forming small nucleoli (Savino et al, 2001; McStay, 2016). They then fuse into larger mature nucleoli, as cells progress through the cell cycle stages in interphase.

To address whether this particular gene positioning of rDNA arrays is affected by the RPL5 depletion, we performed immuno-DNA FISH analyses to detect rDNA arrays and GC (NPM1), simultaneously (Figs 3A and S4D). In control cells (si$Control$), rDNAs were compacted within one to three foci in the nucleolus. In contrast, in RPL5 knockdown cells, the number of rDNA foci increased, and they were scattered throughout the enlarged nucleolus (Fig 3A and B). We further tried to visualize the site of rRNA transcription by 5-ethynyl uridine (EU) incorporation, and the nucleolar UBF, which is located on active rDNA arrays in the nucleolus (Maiser et al, 2020) (Fig S4E). In control cells, UBF was localized as large puncta surrounded by nascent rRNAs. This indicates that active rDNAs are accumulated and bundled together in the nucleolus. In contrast, UBF and EU-labeled nascent rRNAs were in smaller puncta and dispersed in the RPL5-depleted cells. It is noteworthy that the UBF protein levels are comparable in the control and RPL5 knockdown cells (Fig 2B). These results suggested that RPL5 plays a role in gathering the multiple rDNA arrays in the nucleolus, and we reasoned that it is through interactions among RPL5, rDNA and NPM1 (Yu et al, 2006; Mitrea et al, 2016; Yang et al, 2016).

To investigate this hypothesis, we performed a coarse-grained molecular dynamics simulation. Here, the model described the nucleolus by the distributions of four types of particles (Fig 3C): (i) rDNA, the rDNA array that is associated with transcribed rRNA; (ii) NPM1, NPM1 and the associated rRNA; (iii) RPL5; and (iv) a matured ribosome, in which the products are assembled with RPL5 and rRNA. In this model, each rDNA was described by a chain with eight rDNA particles. Based on a recent study (Feric et al, 2016), the affinities between rDNA and NPM1, between two NPM1, and between NPM1 and RPL5 particles were estimated and the affinity between two NPM1 particles was assumed weak, as compared with the others. Each RPL5 particle was assumed to exhibit Brownian motion and be assembled into a matured ribosome, if it was densely surrounded by rDNA or NPM1 (see the Materials and Methods section for the detailed assumptions and the simulation methods of the model).

Our simulation showed the enlarged NPM1 distributions in the absence of RPL5 (Fig 3D and E), which recapitulated our immunofluorescence results (Fig 1E). The NPM1 dispersion levels were 1.34 ± 0.70 $\mu m$ in the control, and 2.08 ± 0.68 $\mu m$ in siRPL5 (Fig S5A), where the dispersion was defined by the average distance of each particle from the center of the masses of rDNA (center of the nucleolus). This result was in good agreement with the enlarged GC region of the nucleolus detected in vivo (Figs 1E and 2A). Importantly, our simulation of steady-state particle positioning revealed more unbundled and scattered distributions of rDNA particles in the siRPL5 model, as compared with the control model (Fig 3D and E and Videos S5 and S6). Quantitative measurements showed that the dispersions of rDNA particles in siRPL5 (1.48 and 0.62 $\mu m$, respectively) were sufficiently larger than those in the control (0.84 and 0.27 $\mu m$, respectively) (Fig 3F), which was also consistent with the rDNA unbundling found in vivo (Fig 3B).

Furthermore, our simulation also gave the time courses of the MSDs of NPM1 particles, which were the power laws of ~0.062 $t^{0.87}$ in

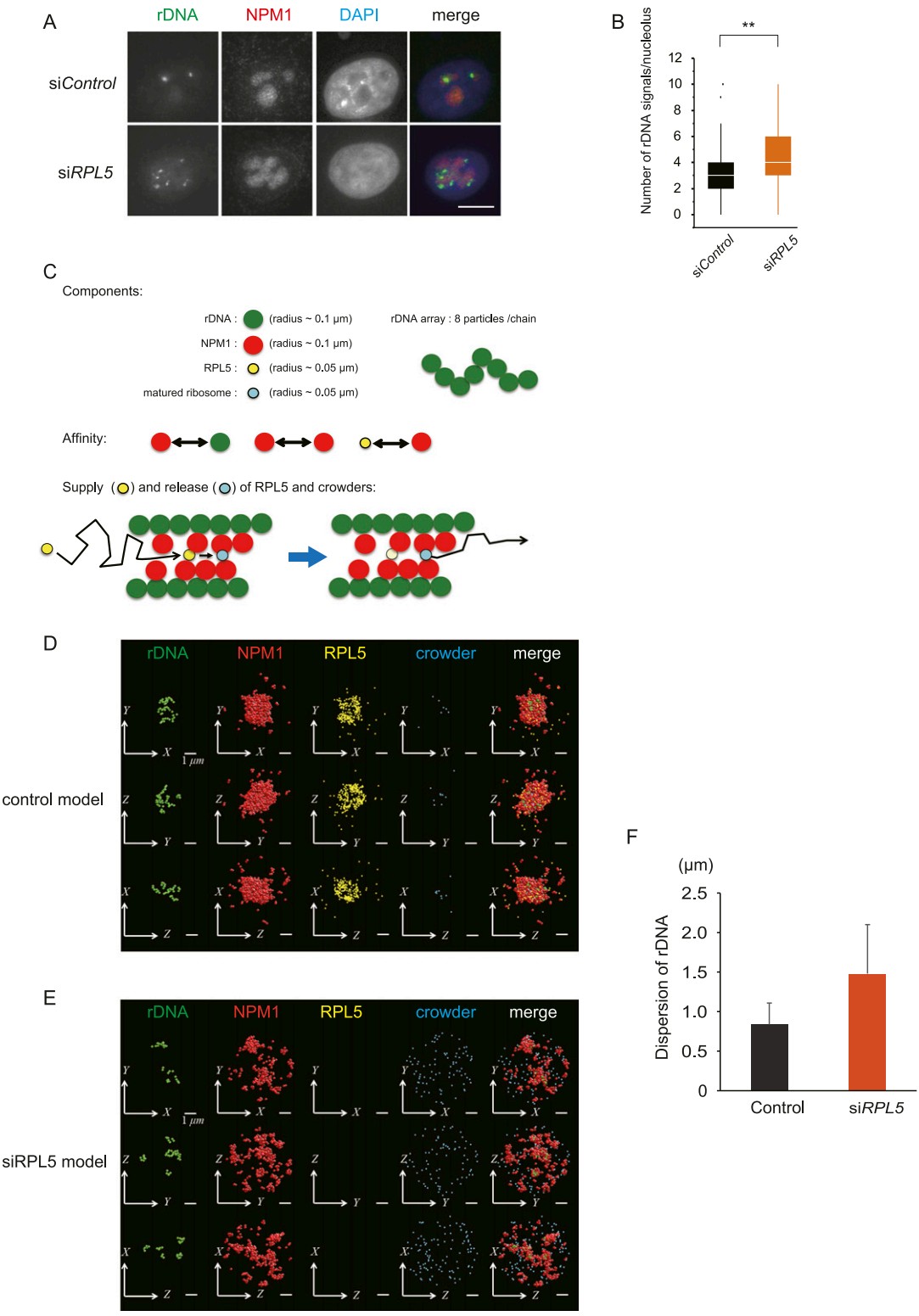

**Figure 3. Loss of RPLs unbundles ribosomal DNA (rDNA) arrays.**
**(A)** Changes of rDNA positioning in RPL5 knockdown cells. Immuno-DNA FISH analyses were performed. rDNA was detected by DNA-FISH (green). The nucleolus and DNA were visualized with NPM1 antibodies (red) and DAPI (blue), respectively. Scale bar, 10 $\mu$m. **(B)** Quantification of immuno-FISH in Fig 3A. **(C)** Coarse-grained model of molecular dynamics in this study. **(D, E)** Simulations of nucleolar formation (rDNA, NPM1, and RPL5) and rDNA position (rDNA) under control (D) and siRPL5 (E) conditions. Scale bar, 1 $\mu$m. RPL5 and matured ribosomes were replaced with virtual crowders in the siRPL conditions (see Materials and methods). **(F)** Dispersions of rDNA measured by the simulations. Values are means ± SD from 2,000 measurements of temporal fluctuations.

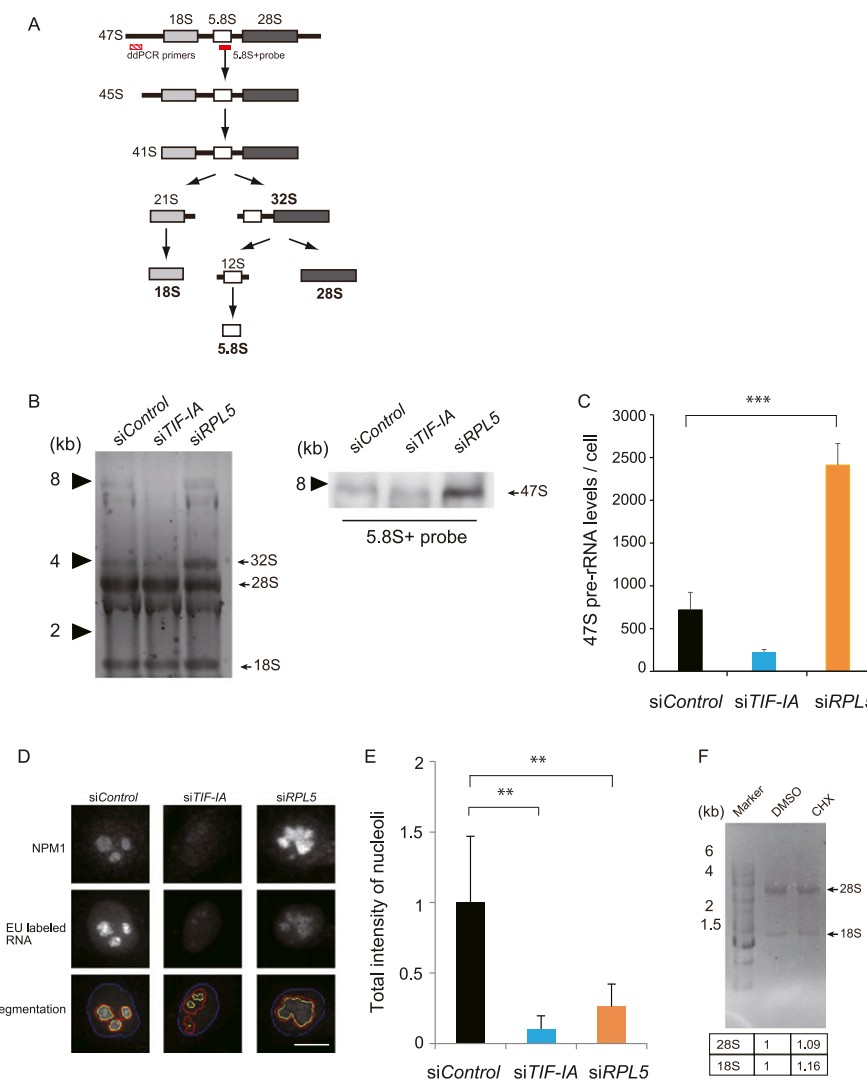

**Figure 4. The RPL5 depletion reduces rRNA transcription and processing.**
**(A)** Scheme of an rRNA gene structure and rRNA processing pathways. The positions of the 5.8S+ probe used for the Northern blot in Fig 4B, and the primer pair used for droplet digital qPCR (ddPCR) in Fig 4C, are shown with red shaded and red solid bars, respectively. **(B, C)** Northern blot and digital PCR analyses showing aberrant accumulation of 47S pre-rRNAs upon RPL5-depletion. **(B)** Equal amounts of RNA from HeLa cells treated with the indicated siRNAs were separated by gel electrophoresis, visualized with GelRed (B, left), and then analyzed by Northern blotting with a 5.8S+ probe (B, right). **(B)** 47S pre-rRNA is indicated (B, right). **(C)** The level of 47S pre-rRNA per single cell was measured by droplet digital PCR. ***$P < 0.01$. **(D)** The newly transcribed rRNA in the nucleus was visualized with an EU incorporation method. Images of EU-labeled RNA and simultaneously NPM1 immuno-labeled nucleoli are indicated. Nuclear and nucleolar regions were segmented with DAPI (blue line) and NPM1 (red line) signals, respectively, together with EU-positive rRNA transcripts within the nucleolus (yellow line). Scale bar, 10 $\mu$m. **(E)** Quantification of rRNA transcription in Fig 4D. rRNA signals in the nucleolus were counted (n > 100 cells), $P < 0.05$; **$P < 0.01$. Values were normalized to that of siControl. **$P < 0.01$. **(F)** Gel electrophoresis of total RNA from HeLa cells treated with DMSO or cycloheximide, CHX. The denaturing gels were stained with ethidium bromide. Quantitative data are shown at the bottom. Values were normalized against those of cells treated with DMSO.

the control model and ~0.099 t$^{0.86}$ in the siRPL5 model: the MSD was larger in the siRPL5 model than in the control model (Fig S5B). This was also consistent with our single molecular trajectory of NPM1 in cells, which was ∝t$^{0.88}$ (Fig 2D). The MSDs of the single FBL molecules that are associated with rDNAs in the control and siRPL5 cells were unchanged and they were ~t$^{0.69}$ (Fig 2D). It is a good agreement with that MSDs of the rDNA particles were ~0.023 t$^{0.69}$ in the control model and ~0.029 t$^{0.68}$ in the siRPL5 model (Fig S5B), where these curves and the power laws to the MSDs seemed quite similar in the control and siRPL5 models. Therefore, the dynamics of NPM1 in cells were well recapitulated with the NPM1 particles, and those of FBL in cells were recapitulated by the rDNA particles in our simulation. More importantly, we suggest that the diffusion of NPM1 molecules is highly restricted by RPL5. These data imply that our simulation recapitulated the molecular dynamics of the nucleolus in cells (Fig 2D). Altogether, we propose that RPL5 is responsible for bundling rDNA arrays through its transient interactions with NPM1 and rDNA while it is in the nucleolus and may promote the rDNA transcription.

## RPL5 contributes to efficient rRNA transcription and processing

The altered nucleolar morphology and the rDNA gene positioning with the RPL5 depletion led us to investigate the rRNA transcription activity. In the nucleolus, a long 47S rRNA precursor is transcribed and processed into three matured rRNAs (18S, 5.8S, and 28S), in a stepwise manner (Fig 4A). We first investigated whether the RPL5 knockdown influences the rRNA processing that normally occurs in the GC region. Using denaturing agarose gel electrophoresis, we visualized the rRNA products, and found that the RPL5 knockdown elicited the abnormal accumulation of the 32S pre-rRNAs (Fig 4B, left). We performed a Northern blot analysis using the 5.8S+ probe, which detects rRNA species (Fig 4A), and found that the 47S pre-rRNA migrating at around 8 kb was increased under the RPL5-depleted conditions (Fig 4B, right), which is consistent with the previous report (Nicolas et al, 2016). Moreover, the analysis by droplet digital PCR (ddPCR) showed that the amount of 47S pre-rRNA per single cell in the steady-state was increased upon the RPL5 depletion (Fig 4C). We concluded that the pre-rRNA was

increased in the RPL5-depleted HeLa cells because of insufficient rRNA processing.

We next tested whether the increased 47S pre-rRNA level is due to the accelerated RNA polymerase I activity in the RPL5 depletion. Accordingly, we measured newly transcribed rRNA by a nucleolar run-on assay with 5′EU incorporation combined with the NPM1 immunofluorescence (Fig 4D and E). We found that the area of RNA synthesis was dispersed throughout the enlarged nucleolus, and the level of RNA synthesis in the nucleolus was decreased in the RPL5 knockdown cells to a level comparable with the TIFI-A knockdown (Fig 4D and E), suggesting that the on-going rRNA transcription is rather decreased in the absence of RPL5. The use of CHX did not affect rRNA processing (Fig 4F). These indicate that the decreased efficiency of rRNA transcription and processing is characteristic of the RPL5 inhibition. This may be due to the unbundled conformation, with a lower density of rDNA arrays and associating factors for transcription and processing.

### Nucleolar deformation is a characteristic of Diamond–Blackfan anemia

DBA is a human congenital red cell aplasia and a founding member of the ribosomopathies. Approximately 50–70% of DBA patients have mutations in RP genes, including *RPL5*. The definitive roles of RPs in DBA pathogenesis are poorly understood. To investigate nucleolar formation in cells from DBA patients and controls, we visualized the nucleolus in peripheral blood lymphocytes by immunofluorescence and analyzed the images with wndchrm to measure the morphological discrepancies (Fig 5). The tested blood cells were obtained from a patient with a heterozygous, large deletion in the *RPL5* gene (Kuramitsu et al, 2012), and showed aberrant nucleolar compartments predominantly at GC (Figs 5A and B and S6A). The dendrogram of NPM1 morphologies indicated that the nucleoli in the cells from the DBA patient were clearly different from those of the controls, and the morphological distance (MD) values from Control_1 were significantly higher in the DBA patient, as compared with Control_2 (Figs 5C and D and S6B). The image features that were useful to discriminate the GCs in the DBA patient and those in the healthy control are summarized in Table S5.

DBA is a bone marrow disorder manifested by impaired erythropoiesis, and thus we confirmed our observations using human leukemic K562 cells, which are able to undergo erythroid differentiation in vitro, and offer a model system for erythroid differentiation in which DBA has a defect (Luo et al, 2017). We knocked down RPL5 in K562 cells, and analyzed their nucleoli by immunofluorescence with anti-NPM1 antibodies (Fig S6C). We found that the nucleolar GC region becomes larger upon the RPL5 depletion, as seen in the patient's lymphocytes and HeLa cells. Because a loss-of-function mutation in RPL5 is frequently found in patients with the congenital defect (Kampen et al, 2020), the nucleolar aberrancy could be transmitted to the whole body (Gazda et al, 2008).

Although the patient lymphocytes and K562 cells are imperfect models, these results indicate that RPL5 plays a nucleolar role in hematopoietic cells and red blood cell precursors may be sensitive to defects caused by reduced levels of RPs during erythropoiesis.

The measurement of nucleolar formation in peripheral blood cells can be a new diagnostic strategy for red cell aplasia.

## Discussion

In this study, we performed high content siRNA screening to identify proteins required for the nucleolar morphology. Among 745 siRNA targets, we identified 16 proteins as being important, including six RPLs. The depletion of RPL5 resulted in aberrant nucleolar morphologies: non-spherical GCs that were unseparated and expanded into the nucleoplasm with unbundled rDNA arrays. We considered the possibility that in normal somatic cells, the local concentration of factors for rRNA transcription and processing increases in the nucleolus because the rDNA arrays are bundled by the interactions of rDNA-NPM1, NPM1-RPL5, and NPM1 themselves (Fig 3). This may facilitate LLPS to create the unique biophysical properties of the nucleolus that are appropriate for efficient rRNA expression (Fig 6A and B).

Our study revealed that specific RPLs, which have long been regarded as simple transient residents of the nucleolus, are actually critically involved in nucleolar assembly and function. Our data suggested that RPL5 functions in spatially positioning and condensing the chromosome sites for rDNA arrays in the nucleolus, and facilitates rRNA transcription and processing.

We showed that RPL5 plays specific structural roles in the nucleolus that are at least partially independent from the general function of protein translation. Our study is based on the following reasons. (1) The mammalian ribosome is composed of the 60S subunit containing 47 RPLs and the 40S subunit containing 33 RPSs, of which 23 RPLs and 11 RPSs have been identified as nucleolar components (Andersen et al, 2005). Among them, we found six RPLs that are critical for the nucleolar morphology. (2) Our quantitative image analysis clearly showed that the nucleolar morphologies in the six RPL knockdowns are similar to each other, and significantly distinct from those in cells treated with a translational inhibitor (Figs 1A–D and S3G and H), excluding the possibility that the disintegrated nucleoli in the RPL knockdowns were simply due to a general defect in translation. (3) Other nuclear domains, including Cajal bodies and nuclear speckles, were intact in RPL-knockdown cells, thus excluding global deterioration effects (Fig S4C). (4) The knockdown of RPL5 reduced rRNA transcription as well as processing, whereas the TIF-IA knockdown only inhibited transcription, and showed a different nucleolar morphology to those of the RPL knockdowns. Furthermore, the TIF-IA knockdown had a milder cell growth defect. (5) Our immunoblots confirmed that the levels of the tested proteins were constant in the control and RPL5 knockdown cells (Fig 2B). This may be due to the abundance of the pre-existing translational machineries, and our RPL5 knockdown conditions might not have been sufficient to influence bulk protein translation.

Previous excellent studies by Lafontaine's group used another image processing algorithm, the iNO scoring method, for an exogenously expressed GFP-fused DFC component, Fibrillarin, as well as an endogenous GC marker, PES1 (Nicolas et al, 2016). Their work revealed that a specific set of RPs is involved in the nucleolar structure, and those that are assembled during the late stage are

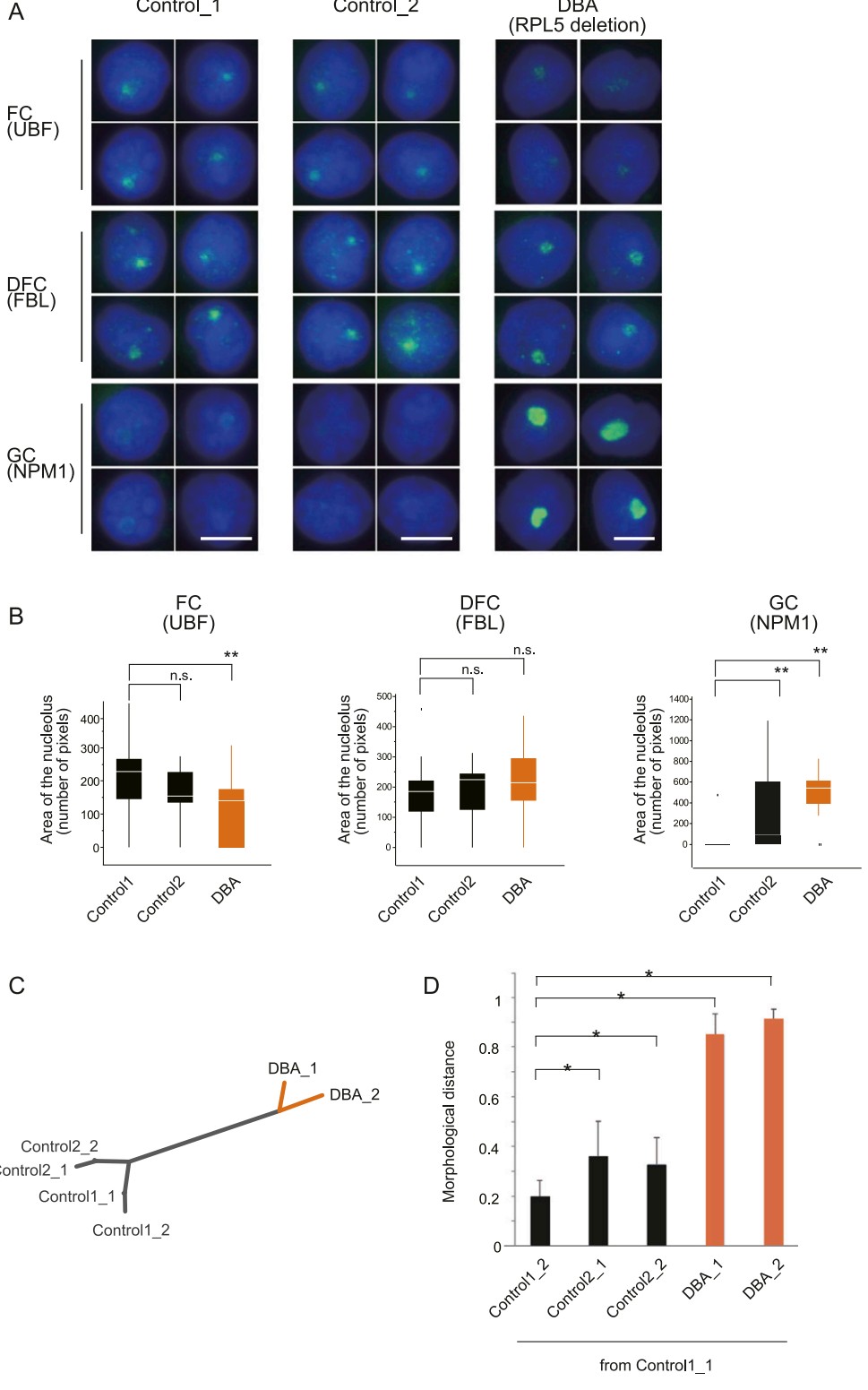

**Figure 5. Nucleolar state is affected in Diamond–Blackfan anemia (DBA).**
**(A)** Analyses of peripheral blood lymphocytes derived from two healthy controls (Control 1 and Control 2) and one patient (DBA). The cells were immunostained to visualize the nucleolar compartments, the FC, DFC, and GC, with the indicated antibodies. Scale bar, 5 $\mu$m. **(B)** Quantification of the nucleolar morphologies in lymphocytes from the DBA patient. Boxes represent the median and the 25% and 75% percentiles, and black dots are outliers. (n = 20) **$P$ < 0.01. **(C)** Dendrogram of nucleolar morphologies in lymphocytes from the two controls (Control 1 and Control 2) and the DBA patient (DBA) (Fig S6A). NPM1 images from each individual were randomly divided into two sub-groups (n = 40), and analyzed with the wndchrm software. **(D)** The MDs from Control1_1, DBA_1, and DBA_2 showed higher distance values, as compared to Control1_2, Control2_1, and Control2_2. This suggests that the nucleolar shapes in the DBA patient were significantly different from those in the healthy individual. Values are the means ± SD of 20 cross validation tests. *$P$ < 0.01.

the most important ones, among the large ribosomal subunits (Nicolas et al, 2016). This previous work used the iNO scoring method, which was specifically developed for nucleolar analyses. In our work, we used wndchrm, which does not require any extra steps specialized for the nucleolar morphologies, including segmentation or special algorithm or pipeline construction. Furthermore, we screened proteins important for the nucleolar morphologies among 745 factors, in which 400 are within the nucleolar proteome,

A    Normal condition

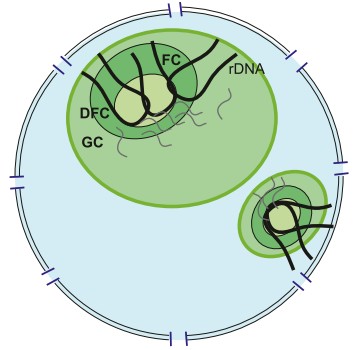

B    RPL5 depletion

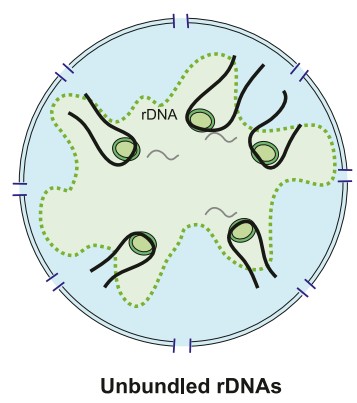

Unbundled rDNAs
↓
Enlarged GC
↓
Defect in rRNA transcription and processing

**Figure 6.   Model for the nucleolar model of RPL5.**
**(A)** Under normal conditions, the ribosomal DNA arrays are bundled and quickly transcribed to rRNAs, which are also efficiently processed. These processes reflect the molecular density of the nucleolus. **(B)** Under RPL5-depleted conditions, the ribosomal DNA arrays are unbundled. As RPL5 normally binds to NPM1 and facilitates phase separation in the GC region, its absence causes the GCs to remain unseparated and enlarged. The GCs lose their sphericity, a hallmark of a liquid droplet, and have lower viscosity. The levels of rRNA transcription and processing are reduced in the altered nucleolus.

and 345 are others including signaling pathways for gene regulation, energy metabolism, and DNA repair. In contrast, the previous work analyzed 80 RPs. Regardless of the fundamentally different experimental set-ups and siRNA targets, both studies resulted in a common finding, and thus solidify the significance of RPL5 as the most important RP for nucleolar structure maintenance. These results also suggested the versatility of the wndchrm image system, which is applicable to other structures in the cell.

In another study, the same group screened factors that are involved in nucleolar structure maintenance among 668 abundant nucleolar proteins, using the iNo scoring method, and identified 86 proteins as being important (Stamatopoulou et al, 2018). Among them, 52 were also in our siRNA library (highlighted in yellow in Table S1). Five of them (RPL5, RPL27A, NPM1, WDR43, and TIF-IA) were among the 16 proteins that we found to be important for nucleolar morphology (Figs 1A and B), again solidifying the importance of RPL5 in the nucleolus.

Furthermore, our study newly discovered that the rDNA arrays were scattered in the RPL5 depletion (Figs 3A and S4D). This could be due to the reduction in interactions among rDNA, RPL5 and NPM1 (Mitrea et al, 2016), as we successfully recapitulated this phenomenon by Blob modeling (Fig 3C–E). This may further lead to the aberrant GC morphology, by the dispersed ribosomal granules and reduced viscosity because of lack of the RPL5–NPM1 interaction, which facilitates the nucleolar assembly through phase separation in vitro (Mitrea et al, 2016). In this altered nucleolar environment, the levels of rRNA transcription and processing are reduced (Fig 4). The significance of the bundled rDNA arrays is not known. However, they may share some properties with the super-enhancers for transcription by RNA polymerase II, which are clusters of enhancers that cooperatively assemble into liquid-like condensates of the transcription factors and mediators to drive robust gene expression (Hnisz et al, 2017; Sabari et al, 2018).

As described above, NPM1 is an assembly facilitator that drives the phase separation of the nucleolus (Mitrea et al, 2016). NPM1 is localized to the nucleolus through multivalent interactions with

proteins containing R-motifs, which are arginine-rich linear motifs. An in vitro experiment revealed that R-motif peptides underwent phase separation with NPM1 (Mitrea et al, 2016). However, it is unlikely that the R-motif itself is the determinant of the nucleolar assembly in somatic cells because only some, but not all, of the RPs contained the R-motif, among the six RPs whose knockdowns resulted in aberrant nucleoli (Fig 1). In addition, the R-motif is present in RPs whose absence did not affect the nucleolar morphology. Therefore, we speculate that the R-motif in RPL5 is used particularly for the nucleolar assembly, and the other five RPLs that were important for the nucleolar morphologies (RPL7, RPL6, RPL28, RPL21, and RPL27A in Fig 1) may contribute to expose the RPL5's R-motif to NPM1 during ribosome assembly in the GC. This is based on the finding that the RPLs are incorporated into adjacent positions in the particle, and their absence could result in steric hindrance of the RPL5–NPM1 interaction (Fig 1D). It may be consistent that the lack of the cognate subunit of RPL5, RPL11, showed a similar effect on nucleolar morphology (Nicolas et al, 2016). RPL5 and RPL11 are functionally related and physically interact in the pre-ribosomal complex (Kressler et al, 2012; Donati et al, 2013).

The depletion of RPL5 resulted in the loss of sphericity and individuality of the nucleolus. We propose that this is due to imperfect phase separation of the GC components, including NPM1. RPL5 may normally contribute to the elasticity of the GC in some way, which then influences the rDNA bundling and viscosity of the nucleolus. Another possibility is that the features of the RPL5 knockdown are caused by the abnormal accumulation of partially assembled ribosomal subunits that are retained in the nucleolus. The enlarged GCs with decreased particle-density may be due to the increased presence of partially assembled ribosomal subunits that interact less with each other. Either way, proper nucleolar properties are necessary for rDNA bundling, rRNA transcription and processing.

In the present study, we showed that nucleolar formation is impaired in a DBA patient with a mutation in an RP gene. The hypertrophy of the GC was significant in lymphocytes derived from a

DBA patient with a heterozygous RPL5 deletion. A previous report showed that rRNA processing is disrupted in DBA-derived cells and that the 32S pre-rRNA accumulates specifically in cells with RPL gene defects (Farrar et al, 2014). Our results provide insight into the molecular mechanism of nucleolar morphology maintenance and suggest a new strategy for disease diagnosis.

# Materials and Methods

### Cell culture and drug treatment

HeLa cells and RPE-1 cells were cultured in DMEM/F12 and DMEM (Sigma-Aldrich), respectively, supplemented with 10% fetal bovine serum, at 37°C in 5% $CO_2$. HeLa cells were treated with cycloheximide (50 $\mu$g/ml) for 6 h for translation inhibition. K562 cells were cultured in RPMI1640 (Sigma-Aldrich) supplemented with 10% fetal bovine serum, at 37°C in 5% $CO_2$.

### siRNA-mediated knockdown

For siRNA screening, HeLa cells cultured in 96-well glass-bottom plates (IWAKI) were treated with siGENOME SMARTpool, which included four unique siRNAs designed for one target gene (Dharmacon). Cells were incubated with siRNA using RNAiMAX (Invitrogen) for 48 h. To evaluate the RNAi knockdown efficiency, each 96-well plate contained a well for a non-targeting siRNA with the sequence CUUACGCUGAGUACUUCGA (GL3, Nippon EGT) as the negative control. The plate also contained a well for the positive control, si*LMNA* (Dharmacon siGENOME SMARTpool: M-004978-01-0005), which was immunostained with anti-LaminA/C under the same conditions used for the other siRNAs. The other SMARTpool siRNAs (Dharmacon) used in this study are as follows:

RPL5: M-013611-01-0005 and L-013611-00-0005, TYMP: M-009281-01-0005, POP4: M-020046-00-0005, RPL9: M-011139-01-0005, RPL13: M-013714-00-0005, RPS9: M-011131-01-0005, and TIF-IA(RRN3): L-016947-00-0005. The individual siRNAs (Dharmacon) used in this study are as follows:

RPL5 #1: GACAAACAGAGAUAUCAUU, RPL5 #2: GAUGAUAGUUCGUGUGACA RPL21 #1: J-012910-07-0002, and RPL21 #2: J-012910-05-0002

### Immunofluorescence and antibodies

Immunofluorescence analyses were performed as previously described (Saitoh et al, 2012). The cells were grown on coverslips and fixed with 4% paraformaldehyde for 15 min at room temperature. For permeabilization, the cells were treated with 0.2% Triton X-100 and 0.5% BSA in PBS for 5 min on ice. For blocking, the cells were washed three times with 0.5% BSA in PBS at room temperature. The cells were incubated with primary antibodies diluted in 0.2% BSA in PBS for 1 h at room temperature. After three washes with PBS, the cells were incubated with secondary antibodies diluted in 0.2% BSA in PBS for 1 h at room temperature. DNA was counterstained with 5′6-diamidino-2-phenylindole (DAPI). To detect the nucleolus, we used three specific antibodies: UBF (sc-13125; Santa Cruz Biotechnology), FBL (ab4566; Abcam), and NPM (sc-6013-R; Santa Cruz Biotechnology).

### Combined immunofluorescence and fluorescence in situ hybridization

HeLa cells were cultured on coverslips and treated with siRNAs for 48 h. The cells were fixed with 4% paraformaldehyde and 0.5% Triton X-100 for 10 min, and then incubated in PBS containing 0.2% Triton X-100 and 0.5% BSA for 5 min on ice for permeabilization. After blocking with 0.5% BSA in PBS for 15 min, the cells were incubated with NPM1 antibodies diluted in 0.2% BSA in PBS for 1 h at room temperature. The cells were washed with PBS, incubated with Cy3-conjugated secondary antibodies for 1 h, and then fixed with 4% paraformaldehyde for 30 min at room temperature. For denaturation, the cells were placed in 2×SSC containing 70% formamide for 3 min at 75°C, and then immediately placed into ice-cold 70% ethanol (Tomita et al, 2015). For dehydration, the cells were washed with 70%, 80%, 95%, and 100% ethanol for 3 min each and air-dried at 42°C.

For the FISH analysis, we used BAC probes covering rDNA (RPCI-11 Human BAC clones; Invitrogen). The BAC probes were labeled with digoxigenin by nick translation (DIG-nick translation mix; Roche), according to the manufacturer's protocol. The probes containing human Cot-1 and yeast tRNA were added to the coverslips and incubated at 37°C overnight. After hybridization, the cells were washed with 2×SSC containing 50% formamide at 37°C for 5 min, followed by 2×SSC (pH 7.0) at 37°C for 5 min. Blocking buffer was added and incubated for 30 min at room temperature. FISH signals were detected with FITC-conjugated anti-digoxigenin. DNA was counterstained with DAPI.

### Image acquisition and quantitative analysis

Images were captured with an IX-71 microscope (Olympus) equipped with a 60× NA1.0 Plan Apo objective lens, a cooled charged-coupled device camera (Hamamatsu), and image acquisition software (Lumina Vision Version 2.4; Mitani Corporation). For FISH, image stacks containing three-dimensional data sets were collected at 0.2-$\mu$m intervals through the z axis, and projected onto two dimensions. For siRNA screening, images were acquired using a Cellomics CellInsight platform (Thermo Fisher Scientific).

The images were statistically analyzed with the HCS studio analysis software (Thermo Fisher Scientific). To recognize the labeled nucleolus, we used the spot detector bioapplication of the HCS studio. First, we identified the DAPI-stained nucleus as the main object, and then the nucleolus in the segmented nucleus was identified as spots. Values of knockdown cells were normalized to that of si*Control*, which was set in every experiment.

For image classification, we used wndchrm ver. 1.31 (Shamir et al, 2010). For machine training, we created image folders composed of multiple images of each knockdown cell; these images were called classes. Each image was pre-excised at 150 × 150 pixels by the Image J1 program (Schneider et al, 2012). In machine training, the folders were used to extract the image features that discriminate classes, and cross validation tests were automatically repeated 20 times. The number of images used for training/test were 64/16 (NPM1 in Fig 1B and C), 32/8 (NPM1 in Fig 5C and D), and 40/10 (NPM1 in Fig S3G and H). The options used for image analysis were a large feature set of 2,873 (-l) and multi-processors (-m). Fisher scores were

automatically computed to discriminate each class. To measure pairwise class similarities, Euclidean distances (d = $\sqrt{\sum(A-B)^2}$) were calculated from the values in the class probability matrices obtained from the cross validations.

## Single molecule imaging

The plasmids encoding SNAP-tagged NPM1 and FBL were generated by cloning the coding sequences of human *Npm1* and *Fbl* into the pSNAPf vector (New England BioLabs) and subcloned into modified pEF5/FRT/V5-DEST (Invitrogen) destination vector in which the EF1α promoter was replaced with human cyclin B1 promoter. HeLa cells stably expressing SNAP-tagged proteins were generated by transfecting the destination vector into HeLa cell line harboring FRT site (pFRT/lacZeo; Invitrogen) using Lipofectamine 2000 (Invitrogen). RPL5 knockdown was performed using BLOCK-iT Pol II miR RNAi Expression Vector Kits (Invitrogen) and the target sequence designed by BLOCK-iT RNAi Designer (Invitrogen, RPL5: 5′-GTTAA GAATAAGGCCTACTTT-3′, non-targeting: 5′-AAATGTACTGCGCGTGGA GAC-3′). The knockdown efficiency was assessed by quantitative RT-PCR as previously described (Lim et al, 2018). HeLa cells expressing SNAP-tagged proteins were transfected with shRNA expression plasmid using Neon Transfection System (Invitrogen) and cultured 24 h, then stained with 3 nM TMR-Star (New England BioLabs) for 5 min and incubated with imaging medium (DMEM high glucose with 10% FBS without phenol red, riboflavin and folic acid) for observation. The knockdown cells were identified by monitoring the expression of EmGFP from shRNA plasmid. The cells were imaged by a HILO fluorescence microscopy set up (Tokunaga et al, 2008) on an inverted microscope (IX-83; Olympus) with a dual C-mount port and a 100× objective (UPlanSApo 100× NA1.40 Oil; Olympus) using HILO and TIRF illuminators (Cell TIRF; Olympus) (Tokunaga et al, 2008). Solid-state lasers (488 nm, 20 mW, Sapphire 488-20-PS, Coherent, Santa Clara, CA, USA, and 561 nm, 50 mW, Compass 561-50, Coherent) were used as light sources for fluorescence excitation. Images were captured simultaneously with two back-thinned electro multiplier charge coupled device cameras (EMCCD, C9100-13; Hamamatsu Photonics). Specimens were observed at 37°C, using a temperature control system with a stage top incubator and an objective heater (IBC-IU2-TOP/-CB/-LH, MI-IBC-IU2; Tokai Hit). Images were recorded with the AQUACOSMOS software (Hamamatsu Photonics) and analyzed using the ImageConverter software (Olympus Software Technology).

## Single-molecule tracking analysis

Single-molecule trajectories were determined using the Particle Tracker plug-in for ImageJ (Sbalzarini & Koumoutsakos, 2005). Trajectories composed of at least 10 steps were used for analysis by calculating the mean square displacements (MSD) (Ito et al, 2017):

$$\rho_k(n\Delta t) = \frac{1}{N_k - n} \sum_{i=1}^{N_i - n} \left( \overrightarrow{r}_{i+n} - \overrightarrow{r}_i \right)^2, \ n = 1, ..., N_k - 1, \qquad (1)$$

where $\rho_k(n\Delta t)$ is the MSD of duration $n\Delta t$, $\Delta t$ is the frame interval (33.33 ms), $N_k$ is the number of spots on the $k$-th trajectory ($k = 1, ..., N_{traj}$), $N_{traj}$ is the number of trajectories, and $r_i = (x_i, y_i)$ is the position

of the $i^{th}$ spot. The averaged MSD $\overline{\rho}(n\Delta t)$ was calculated by averaging $\rho_k(n\Delta t)$ for all of the trajectories ($k = 1, ..., N_{traj}$) obtained from images of 20 cells observed on the same day. The diffusion coefficient $D$ was determined by fitting the averaged MSD $\overline{\rho}(t)$ with the following equation of the confined diffusion (Feric & Brangwynne, 2013):

$$\rho(t) = 4Dt^\alpha, \qquad (2)$$

where $D$ is the diffusion coefficient, and $\alpha$ is a diffusive exponent, which takes values of 1 for simple diffusions, <1 for confined diffusions, and >1 for directional diffusions. Three data sets of $D$ and $\alpha$ were obtained from experiments on different days, and the averages of the three data sets were used as the final results (Table S4). The viscosity $\eta$ was estimated using the following Stokes–Einstein equation for a sphere:

$$D = \frac{k_B T}{6\pi\eta r}, \qquad (3)$$

where $k_B$ is Boltzmann's constant, $T$ is the absolute temperature, and $r$ is the radius of the sphere.

## RNA extraction and electrophoresis

Total RNA was isolated using the TRIzol Reagent (Ambion), according to the manufacturer's protocol. RNA was treated with DNase I (Roche) at 37°C for 20 min. To evaluate rRNA processing, total RNA (2 µg) was denatured at 65°C for 15 min and separated on denaturing agarose gels (18% formaldehyde and 1.2% agarose in MOPS buffer) by electrophoresis at 4 V/cm. After electrophoresis, the RNA concentration was quantified by the ImageQuant TL software (GE Healthcare).

## Quantitative RT-PCR and primer sequences

Total RNA was converted to cDNA, using a High Capacity cDNA Reverse Transcription Kit (Applied Biosystems). Quantification of target cDNAs was performed using SYBR Green PCR Master Mix (Toyobo). Values were normalized against β-actin expression before calculating the relative fold changes. The following primers were used:

β-actin, 5′-CCAACCGCGAGAAGATGA-3′, 5′-CCAGAGGCGTACAGGGAT AG-3′; RPL5, 5′-CACTGGCAATAAAGTTTTTGGTG-3′, 5′-AACCAGGGAAT CGTTTGGTA-3′; TYMP; 5′-GCACCTTGGATAAGCTGGAG-3′, 5′-CTCTGAC CCACGATACAGCA-3′, TIF-IA; 5′-AACATCTTTGACAAACTCCTGTTG-3′, 5′-AAAATGCCTCTGCGAATCC-3′, POP4; 5′-TGCGGCTCTTTGACATTAAAC-3′, 5′-TCATGGAGAGGGAGGAAAAG-3′, RPL21; 5′-GAGCCGAGATAGCTTCCT GA-3′, 5′-CTCCTTCCCATTGGTTCTCA-3′, RPL9; 5′-ACGTTCTTTCTTTGCT GCGT-3′, 5′-CCCTTCACGATAACTGTGCG-3′, RPL13; 5′-GCCTTCGCTAGT CTCCGTAT-3′, 5′-ACTGATTCCAAGTCCCCAGG-3′, RPS9; 5′-GCTGACGCT TGATGAGAAGG-3′, 5′-CCTCTATCTTCAGGCCCAGG-3′.

## Immunoblot analysis

The cells were disrupted in SDS sample buffer containing benzonase (Sigma-Aldrich). The total protein was separated by SDS-PAGE and transferred to a nitrocellulose membrane (GE Healthcare), and then analyzed by immunoblotting using Western Lightning Plus-ECL (Perkin Elmer). Protein levels were quantified using the ImageQuant

TL software (GE Healthcare). Primary antibodies for immunoblot used in this study were as follows: RPL5 (GTX101821; Gene Tex), RPL21 (A305-032A; Bethyl Laboratories), H3 (MABI0301; MBL Life Science), p53 (sc-126; Santa Cruz Biotechnology), and p21 (sc-6246; Santa Cruz Biotechnology).

## Detection of rRNA transcription

Nascent RNA was detected using a 5-ethynyl uridine (EU) imaging assay kit (Invitrogen), according to the manufacturer's protocol. Briefly, HeLa cells were cultured on coverslips in 12-well plates. 48 h after the siRNA transfection, 0.5 M EU was added to the cells and incubated at 37°C for 15 min. The cells were fixed with 4% para-formaldehyde and the RNA labeled by EU was visualized by im-munofluorescence. Simultaneously, the cells were immunostained with NPM1 antibodies to detect nucleoli. To quantify rRNA tran-scription, we measured the intensity of the labeled RNA that coin-cided with the NPM1 signals, using the HCS studio software (Thermo Fisher Scientific). Values were normalized to that of siControl.

## Electron microscopy

Electron microscopy was performed as described previously (Haraguchi et al, 2008, 2015). HeLa cells were cultured in glass-bottom dishes (MatTek) and fixed with glutaraldehyde at a final concentration of 2.5% for 1 h. Cells were post-fixed with 1% $OsO_4$ (Nisshin EM) in phosphate buffer, pH 7.4, for 1 h, washed briefly with distilled water, and then stained with 2% uranyl acetate (Wako) for 1 h. The sample was sequentially dehydrated with 30%, 50%, 70%, 90%, and 100% ethanol, and embedded in epoxy resin by sequential incubations with 10%, 30%, 50%, and 70% (V/V) Epon812 (TAAB) in ethanol for 3 min, 90% for 10 min, and 100% Epon812 for 1 h. The sample was further incubated with 100% Epon812 overnight and for another 3 h. To make the epoxy block, the sample was incubated with 100% Epon812 at 60°C for 48 h. The epoxy block was trimmed, and ultra-thin sections with an 80 nm thickness were cut using an ultramicrotome (Leica Microsystems). Thin sections were stained with 4% uranyl acetate (Wako) for 15 min and a commercial ready-to-use solution of lead citrate (Sigma-Aldrich) for 1 min. Image data were collected with a JEM1400 electron microscope (JEOL) at 80 kV.

## Coarse-grained molecular dynamics model

The coarse-grained molecular dynamics model of the nucleolus was developed using four types of particles: (i) rDNA particle that de-scribes a part of the rDNA array attached to its transcripts, rRNAs, (ii) NPM1 particle that describes the high density populations of NPM1 and rRNA molecules, (iii) RPL5 particle, and (iv) matured-ribosome particle that describes the high density populations of biochemical products of molecular complexes formed by RPL5 and rRNA.

Each rDNA in the rDNA arrays was assumed to be a chain of rDNA particles. The affinities between rDNA and NPM1 particles, between the pair of NPM1 particles, and between NPM1 and RPL5 particles were assumed. Notably, the former two affinities occur indirectly, because of the rRNA-FC affinity and rRNA-NPM1 affinity (Feric et al, 2016). The latter affinity is due to the direct affinity between NPM1 and RPL5 molecules (Mitrea et al, 2016).

The motion of each particle was assumed to obey the following Langevin equations:

$$\gamma_i \frac{d\mathbf{x}_i}{dt} = -\nabla_i V + \boldsymbol{\eta}_i(t), \tag{4}$$

$$\boldsymbol{\eta}_i(t)\boldsymbol{\eta}_j(t') = 6\gamma_i Q \delta_{ij}\delta(t-t'), \tag{5}$$

$$\gamma_i = 6\pi\mu_{nuc} r_i, \tag{6}$$

where $i$ indicates the particle index and $\mathbf{x}_i$ indicates the three-dimensional position vector of the $i$-th particle $\mathbf{x}_i = x_i\mathbf{e_x} + y_i\mathbf{e_y} + z_i\mathbf{e_z} = (x_i, y_i, z_i)$, in which $\mathbf{e}_x$, $\mathbf{e}_y$, and $\mathbf{e}_z$ are unit vectors in the directions of the $x$, $y$, and $z$ axes in Cartesian coordinate space. $\gamma_i$, $\boldsymbol{\eta}_i(t)$, and $r_i$ indicate the drag coefficient for the $i$-th particle from the nucle-oplasm, the random force working on the $i$-th particle, and the radius of the $i$-th particle. The function $V$ gives the potential of the mechanical forces among particles, and $\nabla_i = \mathbf{e}_x\frac{\partial}{\partial x_i} + \mathbf{e}_y\frac{\partial}{\partial y_i} + \mathbf{e}_z\frac{\partial}{\partial z_i}$. The quantity $Q$ gives the strength of the random force, and $\delta_{ij}$ and $\delta(t-t')$ are the Kronecker's delta and Dirac's delta functions.

The viscosity of the nucleoplasm was assumed as $\mu_{nuc}$ = 0.64 ($kg\ m^{-1}sec^{-1}$) (Lin et al, 2015) and the radii of particles were empirically assumed as $r_i$ = 1.0 × 10$^{-7}$ $m$ for rDNA and NPM1 particles, and $r_i$ = 0.5 × 10$^{-7}$ $m$ for RPL5 particles and the matured ribosome particles. In the present simulations, $Q$ was assumed to obey $Q/\gamma_i$ = 1 × 10$^{-14}$ ($kg\ m^2sec^{-1}$) for rDNA and NPM1 particles ($Q/\gamma_i$ = 2 × 10$^{-14}$ ($kg\ m^2sec^{-1}$) for RPL5 and matured-ribosome particles), with which the order of $Q$ was the same as $k_BT$ with $T \sim 300\ K$.

The potential of the mechanical forces among particles, $V$, was assumed as follows:

$$V = V_{chain} + V_{FM} + V_{MM} + V_{MR} + V_{ex}. \tag{7}$$

Here, $V_{chain}$ is the potential energy function for the bonded in-teraction between the neighboring rDNA particles to form the rDNA particle chains. The functions $V_{FM}$, $V_{MM}$, and $V_{MR}$ are the potentials providing the affinities between the rDNA and NPM1 particles, be-tween two NPM1 particles, and between the NPM1 and RPL5 particles, respectively. The function $V_{ex}$ is the hard-sphere potential between two particles, except for the abovementioned pairs of particle types that represented the excluded volume of each particle.

The potential function $V_{chain}$ is given by the following equation:

$$V_{chain} = \sum_{i<j}\frac{k_c}{2}\left(|\mathbf{x}_i - \mathbf{x}_j| - (r_i + r_j)\right)^2, \tag{8}$$

where the $i$-th particle is a part of the rDNA particle chain and Σ indicates the sum of the $i$- and $j$-th particles ($i < j$) that are adjacent and bonded along the rDNA.

The potential function $V_{FM}$ is given by the following equation:

$$V_{FM} = \sum_{i<j}\theta\left((r_i + r_j) - |\mathbf{x}_i - \mathbf{x}_j|\right) \times \frac{k_{FM}^{ij}}{2}\left[\left(|\mathbf{x}_i - \mathbf{x}_j| - 0.88(r_i + r_j)\right)^2 - \left(0.12(r_i + r_j)\right)^2\right], \tag{9}$$

where the $i$-th particle is an rDNA or NPM1 particle and Σ indicates the sum of the $i$- and $j$-th particles ($i < j$) in the case where the $i$-th

($j$-th) particle is one of the rDNA particles and the $j$-th ($i$-th) particle is one of the NPM1 particles. The function $\theta$ is a Heaviside step function, and the coefficient $k_{FM}^{ij}$ correlates to the strength of the affinity between these particles. Similarly, the function $V_{MM}$ is given by the following equation:

$$V_{MM} = \sum_{i<j}\theta\big((r_i + r_j) - |\mathbf{x}_i - \mathbf{x}_j|\big) \times \frac{k_{MM}^{ij}}{2}\Big[\big(|\mathbf{x}_i - \mathbf{x}_j| - 0.91(r_i + r_j)\big)^2$$
$$- \big(0.09(r_i + r_j)\big)^2\Big], \tag{10}$$

where the $i$-th particle is NPM1 and $\Sigma$ indicates the sum of the $i$- and $j$-th particles ($i < j$) in the case where both the $i$- and $j$-th particles are NPM1. The coefficient $k_{MM}^{ij}$ correlates to the strength of the affinity between these particles. The function $V_{MR}$ is also given in a similar form as follows:

$$V_{MR} = \sum_{i<j}\theta\big((r_i + r_j) - |\mathbf{x}_i - \mathbf{x}_j|\big) \times \frac{k_{MR}}{2}\Big[\big(|\mathbf{x}_i - \mathbf{x}_j| - 0.92(r_i + r_j)\big)^2$$
$$- \big(0.08(r_i + r_j)\big)^2\Big], \tag{11}$$

where the $i$-th particle is NPM1 or RPL5 and $\Sigma$ indicates the sum of the $i$- and $j$-th particles ($i < j$) in the case where the $i$-th ($j$-th) particle is one of the NPM1 particles and the $j$-th ($i$-th) particle is one of the RPL5 particles. The coefficient $k_{MR}$ correlates to the strength of the affinity between these particles.

The function $V_{ex}$ is given as follows:

$$V_{ex} = \sum_{i<j}\theta\big((r_i + r_j) - |\mathbf{x}_i - \mathbf{x}_j|\big) \times \frac{k_{ex}}{2}\big(|\mathbf{x}_i - \mathbf{x}_j| - (r_i + r_j)\big)^2, \tag{12}$$

where $\Sigma$ indicates the sum of all pairs of $i$- and $j$-th particles ($i < j$), except for the particle pair that the forces by the potentials $V_{chain}$, $V_{FM}$, $V_{MM}$, and $V_{MR}$ work on.

Note that $V_{FM}$, $V_{MM}$, and $V_{MR}$ involve minimum values at $|\mathbf{x}_i - \mathbf{x}_j| = 0.88 (r_i + r_j)$, $|\mathbf{x}_i - \mathbf{x}_j| = 0.91 (r_i + r_j)$, and $|\mathbf{x}_i - \mathbf{x}_j| = 0.92 (r_i + r_j)$, respectively. This represents the fact that these potentials describe the affinities between particles.

The coefficients $k_{FM}^{ij}$ and $k_{MM}^{ij}$ are assumed as follows:

$$k_{FM}^{ij} = \frac{1}{2}\left(\frac{1}{c_i} + \frac{1}{c_j}\right)k_{FM}^0, \tag{13}$$

$$k_{MM}^{ij} = \frac{1}{2}\left(\frac{1}{c_i} + \frac{1}{c_j}\right)k_{MM}^0. \tag{14}$$

Here, $c_i$ is the number of contacting NPM1s if the $i$-th particle is rDNA, and that is the number of contacting rDNA and NPM1 if the $i$-th particle is NPM1; we considered two particles to be in contact when the distance between them was closer than the sum of their radii. Equations (13) and (14) indicate that the affinities from each NPM1 and each rDNA to other NPM1 particles are assumed to decrease with $c_i$. This assumption is based on the following considerations.

rDNA and NPM1 particles are assumed to describe the molecular populations containing rRNAs. Here, the affinity between two such molecular populations was naturally considered to increase the

sum of the number of rRNAs on their contacting surfaces. Notably, for each molecular population, the number of rRNAs per one contacting surface with another molecular population decreases with the increase in the number of contacts with other molecular populations, indicating that the affinity from one particle to other particles seems to decrease with the number of contacting particles.

In the present model, the parameters in the potential function, $k_{chain}$, $k_{FM}^0$, $k_{MM}^0$, $k_{MR}$, $k_{ex}$, and $k_b$ were assumed as obeying $k_{chain}r^2 = k_{FM}^0 r^2 = k_{MM}^0 r^2 = k_{MR}r^2 = k_{ex}r^2 = k_b r^2 = 256Q$ with $r = 10^{-7}\, m$. Here, the maximum heights of the potential barriers of $V_{FM}$, $V_{MM}$, and $V_{MR}$ to escape the attractive forces among particles in the simulations were given as ~$7.4Q$, ~$4.14Q$, and ~$1.85Q$, respectively.

For the simulations in Figs 3C–F and S5, an RPL5 particle was assumed to transform into a matured ribosome once every 10 s on average, when the sum of the rDNA or NPM1 particles that were within $4 \times 10^{-7}\, m$ from this RPL5 was more than 48.

To supply RPL5 in the simulation of the control model, we artificially replace a matured ribosome with an RPL5 particle once every 10 s on average, when no particles exist in the region within $4 \times 10^{-7}\, m$ from this particle. On the other hand, the siRPL model contained the crowders, instead of RPL5 and matured ribosomes, to make the volume fractions of particles in the siRPL5 model equal to those of the control model. Each crowder was assumed to have the same physical properties as a matured ribosome.

### Simulation and analysis of coarse-grained model

To consider the behaviors of the nucleolus, we used the above-mentioned model with 10 rDNA chains, 1,456 NPM1 particles, and 512 particles that were assumed to be RPL5 or matured ribosome particles in the control model, and 512 particles that were assumed to be virtual crowders in the siRPL5 model, where each rDNA chain was composed of eight rDNA particles. For the simulation, the time integral of Langevin Eq. (M1) was calculated numerically using the Eular–Maruyama method (Kloeden & Platen, 1992) with a unit MD step = $2^{-15}$ s. We used the restriction that all particles could migrate in the spherical region within the radius = 3.2 $\mu m$ from the nucleolus center, for the simplicity and convenience of the simulations.

The MSD of NPM1 particles was calculated as the average of the square displacements of NPM1 particles, except those with sizes within the top and bottom 5%, to eliminate the influences of outliers. The MSD of rDNA particles was calculated using all rDNA particles.

### Three-dimensional models of a human ribosome particle

The models of the human ribosome structure in Fig 1D were created using the atomic coordinates previously deposited in the RCSB Protein Data Bank, with the code 4V6X, and the PyMOL software (The PyMOL Molecular Graphics System, Version 2.0 Schrödinger, LLC).

### Northern blot analysis

At 48 h following the treatment with siRNA, total RNAs were extracted from the cells using TRI REAGENT (Molecular Research Center, Inc.). Purified total RNAs (4 $\mu g$) were denatured and

separated by 0.8% formaldehyde-agarose gel electrophoresis, and the gel was stained with GelRed Nucleic Acid Gel Stain (Biotium, Inc.). The fractionated RNAs were transferred and cross-linked to an Amersham Hybond-N+ membrane (Cytiva), which was used for prehybridization and hybridization with the DIG-labeled 5.8S+ probe (Sirri et al, 2016) using DIG Easy Hyb buffer (Roche) at 50°C. After washing, the membrane was incubated with the DIG Wash and Block Buffer Set (Roche) and alkaline phosphatase–conjugated anti-digoxigenin antibodies (Roche). Detection of alkaline phosphatase–labeled 47S pre-rRNA was performed using the chemiluminescent substrate CDP-Star (Roche).

### Droplet digital PCR

HeLa cells were transfected with control, TIF-IA or RPL5 siRNA and incubated for 48 h. The total cell numbers were calculated using an automated cell counting device (CountessII FL; Thermo Fisher Scientific), and total RNAs were subsequently purified using an RNeasy Mini Kit (QIAGEN), according to the manufacturer's instructions. cDNA was synthesized by reverse transcription (ReverTra Ace qPCR RT Master Mix, Toyobo) using template RNA equivalent to 2,500 cells. After sample dilution, ddPCR was performed using primer pairs for 47S pre-rRNA (5′-TGTCAGGCGTTCTCGTCTC, 5′-CAC-CACATCGATCACGAAGA), and the copy number of 47S per cell was measured using the QX200 Droplet Digital PCR System (Bio-Rad).

### Patient samples

Patient blood smears were provided by the Department of Paediatrics, Hirosaki University Graduate School of Medicine. Informed consent was obtained from patients. This study was approved by the Ethics Committees of Hirosaki University Graduate School of Medicine and Kumamoto University Graduate School of Medicine.

### Statistical analysis

Differences between groups were analyzed using the $t$ test. $P < 0.05$ was considered statistically significant.

# Supplementary Information

# Acknowledgements

We thank Chiyomi Sakamoto and Yoko Yasuda for technical assistance and all members of the Nakao and Saitoh laboratories for their support. We thank Jun Katahira (Osaka University) for discussions and technical assistance; Keiji Kimura (Tsukuba University), Masahiko Harata (Tohoku University), Makoto Tsuneoka (Takasaki University of Health and Welfare), Miki Hieda (Ehime Prefectural University of Health and Sciences), and Christian Lanctôt (Charles University) for valuable discussions and advice; and Yoshihiro Koyama (Thermo Fisher Scientific) for technical support. This work was supported by JSPS KAKENHI Grant Numbers JP19K23927 and JP20K07578 (to K Watanabe); JP20H05397 and JP20K06496 (to H Tachiwana); JP18H05528 (to T Haraguchi); JP18K14661, JP18H05527, and JP20K15755 (to Y Ito); JP 25116007 and JP 19H03192 (to M Tokunaga); 18H02618 (to M Nakao); JP18H05531, JP18K19310, and JP20H0352 (to N Saitoh); by grants from the Takeda Science Foundation (to N Saitoh); The Vehicle Racing Commemorative Foundation (to N Saitoh); a Research Grant of the Princess Takamatsu Cancer Research Fund (to N Saitoh), and the Japan Agency for Medical Research and Development (CREST) (to M Nakao). K Watanabe and N Saitoh are supported by a research grant from DAIZ, Inc. H Matsumori is supported by a Young Scientist Fellowship from the Japan Society for the Promotion of Science (JSPS). IG Goldberg is supported by the Intramural Research Program of the U.S. National Institutes of Health, National Institute on Aging.

## Author Contributions

H Matsumori: conceptualization, formal analysis, supervision, funding acquisition, and investigation.
K Watanabe: formal analysis, funding acquisition, and investigation.
H Tachiwana: formal analysis, funding acquisition, and investigation.
T Fujita: investigation and methodology.
Y Ito: funding acquisition, investigation, and methodology.
M Tokunaga: funding acquisition, investigation, and methodology.
K Sakata-Sogawa: investigation and methodology.
H Osakada: investigation.
T Haraguchi: investigation and methodology.
A Awazu: methodology.
H Ochiai: funding acquisition and investigation.
Y Sakata: investigation.
K Ochiai: resources.
T Toki: resources.
E Ito: resources and methodology.
IG Goldberg: investigation and methodology.
K Tokunaga: supervision, funding acquisition, and investigation.
M Nakao: supervision, funding acquisition, investigation, and methodology.
N Saitoh: conceptualization, resources, supervision, funding acquisition, and investigation.

## Conflict of Interest Statement

The authors declare no competing financial interests. DAIZ Inc. had no control over the interpretation, writing, or publication of this work.

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
