## [Reviewer comments · Life Science Alliance]

Life Science Alliance

Ribosomal protein L5 facilitates rDNA-bundled condensate and nucleolar assembly

Haruka Matsumori, Kenji Watanabe, Hiroaki Tachiwana, Tomoko Fujita, Yuma Ito, Makio Tokunaga, Kumiko Sakata-Sogawa, Hiroko Osakada, Tokuko Haraguchi, Akinori Awazu, Hiroshi Ochiai, Yuka Sakata, Koji Ochiai, Tsutomu Toki, Etsuro Ito, Ilya Goldberg, Kazuaki Tokunaga, Mitsuyoshi Nakao, and Noriko Saitoh

DOI: <https://doi.org/10.26508/lsa.202101045>

Corresponding author(s): Noriko Saitoh, Japanese Foundation For Cancer Research and Mitsuyoshi Nakao, Kumamoto University

Review Timeline:

Submission Date:	2021-02-03
Editorial Decision:	2021-04-07
Revision Received:	2022-01-17
Editorial Decision:	2022-02-21
Revision Received:	2022-03-01
Accepted:	2022-03-02

Transaction Report:

April 7, 2021

Re: Life Science Alliance manuscript #LSA-2021-01045-T

Noriko Saitoh
The Cancer Institute of JFCR
JAPAN

Dear Dr. Saitoh,

Thank you for submitting your manuscript entitled "Ribosomal protein L5 facilitates rDNA-bundled condensate and nucleolar assembly" to Life Science Alliance (LSA). The manuscript was assessed by expert reviewers, whose comments are appended to this letter.

As you will note from the reviewers' comments below, the reviewers are intrigued by these findings, but have raised a number of concerns that need to be addressed prior to further consideration of the manuscript at LSA. While the reviewers' requests are somewhat extensive, we do think that addressing these will strengthen the manuscript further, and are willing to extend the revision timeframe longer than the typical 3 months. Please let us know if you think you will need more time than 3 months to revise this study.

To upload the revised version of your manuscript, please log in to your account: <https://lsa.msubmit.net/cgi-bin/main.plex>. You will be guided to complete the submission of your revised manuscript and to fill in all necessary information. Please get in touch in case you do not know or remember your login name.

Please note that papers are generally considered through only one revision cycle, so strong support from the referees on the revised version is needed for acceptance.

Thank you for this interesting contribution to Life Science Alliance. We are looking forward to receiving your revised manuscript.

Sincerely,

Shachi Bhatt, Ph.D.
Executive Editor
Life Science Alliance
<http://www.lsajournal.org>
Tweet @SciBhatt @LSAjournal

B. MANUSCRIPT ORGANIZATION AND FORMATTING:

Reviewer #1 (Comments to the Authors (Required)):

The authors conducted RNAi screen of 745 human genes that were previously associated with nucleolus, in search for genes involved in nucleolar structure. They identified 16 genes essential for normal distribution of nucleophosmin as proxy of nucleolar organisation. Image analysis with wncdhrm was employed to classify the phenotype resulting from the RNAi of the 16 genes, not the 745 genes. The authors decided to focus on characterisation of RPL5, which led the authors conclude that RPL5 has roles in nucleolar organisation. A similar, but more extensive study was published by Nicolas et al. in 2015 (cited in the manuscript as ref 21), including roles of RPL5 and other ribosomal proteins in nucleolar integrity.

Specific points

The initial RNAi screen is not conducted carefully. The depletion of mRNA for each RNAi should be tested in order to evaluate the phenotype.

p6 lines 6-7

Fig S1b shows data on intensity and area of NPM1 signals - not morphology.

Fig S1b

What are the data distribution? The 16 selected genes do not appear to show any particular features in these data set. It seems that the selection criteria is independent of quantification in Fig S1b.

p6 line 7-8

What are the true criteria for selection of the 16 genes?

Table S1 and Fig S1b

It is more useful if the data is sorted by rank or in alphabetical order, but not the index number. It was difficult to find genes of my interest e.g. RPL11 or NCL from the list.

p7 line 2 / Fig S3a

Are their mRNA expression reduced as much as siRPL5?

Fig S3b

Please show the area of nucleolus.

Fig 1c

What are the morphological distances for other RPL, e.g. RPL9 and RPL13?

P6 lines 17-21

Which parameters were selected to define groups and draw the dendrogram?

Discrepancy between Fig S3g and Fig 1b

siRPL5 and siRPL21 look similar in Fig3g, but they are mapped on different branches and siControl is mapped in between in Fig 1b. Are the parameters to calculate the distance same in Fig S3g and Fig 1b?

Fig 2a is not compelling evidence for RPL5 as a LLPS facilitator.

Fig S3d

The intensity for histone H3 is not shown.

Fig S3e

Growth curves are easier to interpret if drawn in log scale.

Fig S4

siControl cells passing through mitosis are more appropriate control to show that aberrant nucleoli are phenotype attributed to siRPL5, not a cell cycle event.

I am surprised with notable absence of reference to RPL11 - a well known cognate subunit of RPL5.

Reviewer #2 (Comments to the Authors (Required)):

In this work, the authors used siRNA screening combined with machine learning to find genes whose knockdown affects the morphology of the nucleolus. The screen was conducted in HeLa cells using nucleolar protein NPM1 as a readout. Data were classified with an open-source software Wndchrm. Hits included a group of RPL proteins that are integral components of the ribosome. Their knockdowns shared common morphological features, most notably enlarged size and loss of sphericity of the nucleolus.

Authors attribute RPL5 knockdown to imperfect phase separation, speculating that RPL5 may contribute to the elasticity and surface tension of the Granular Component (GC) of the nucleolus, and can influence its viscosity that was measured by single-molecule tracking of SNAP-tagged NPM1. Authors propose that RPL5 facilitates interactions between rDNA, itself, and NPM1, thereby promoting liquid-liquid phase separation and expression of rRNA. In the end, the authors explored nucleolar features in lymphocytes from a patient(s) suffering from Diamond Blackfan anemia (DBA).

Many experiments in this paper appear to be conclusive and well-done, such as electron microscopy and single-particle tracking analysis. In my opinion LLPS is still at the hypothesis stage for nucleoli, and not a foregone conclusion for how they organize. I don't see why the observed features of several RPL knockdowns must be attributed specifically to LLPS and cannot be caused simply by abnormal accumulation of partially assembled ribosomal subunits that are not fully assembled due to lack of key subunits, and are retained in the nucleolus. The granular component is the compartment where the last stages of assembly of pre-ribosomal particles occur, preceding their transport out of the nucleolus. It is plausible that in the absence of given RPLs this enlarged GC with decreased density may be filled with more unassembled or partially assembled ribosomal subunits and fewer assembled pre-ribosomal particles. Abnormal accumulation of unassembled proteins can also impair overall nucleolar function, including transcription. To further test the involvement of RPLs in LLPS the authors could include in vitro experiments with isolated components, but I think that is outside the scope of the current manuscript. Previous in vitro studies with NPM1 showed that it can undergo homotypic and heterotypic LLPS at sufficient concentrations in systems that did not contain RPLs. Also, more studies are needed to show that LLPS directly facilitates rDNA transcription by PolI in vivo. My suggestion to authors is to be wary of interpreting current results solely from the perspective of LLPS. At a minimum they should provide an alternative explanation of their findings that does not invoke LLPS.

It would be significantly strengthen the authors findings to confirm the RPL knockdown phenotype in cell lines other than HeLa. HeLa cells are highly aneuploid and adapted to constant protein dosage imbalance, and their p53 pathway is not functional. Besides, nucleolar size and shape in different cell lines can vary. It would significantly strengthen the manuscript to confirm the RP knockdowns in HeLa cells, and confirm/extend the knockdowns and nucleolar phenotype in a minimum of one additional cell line that is karyotypically normal.

Other points:

1. The quality of EM images in figure 2a is impressive and provides strong evidence for decreased GC density in RPL5 knockdown. This can be interpreted as a decreased density of mature ribosomal subunits that appear as small dark granules on TEM labeled with uranyl acetate. I do not see the triangles referred to in the figure legend.
2. Only Rpl5 knockdown is confirmed by Western. Some journals require that phenotypes are observed with at least two unique siRNAs to avoid reporting off-target effects. In this study a smart pool was used with 4 separate siRNAs. The authors should consider confirming their hits (at least RP) with individual siRNAs and corresponding Western blotting.
3. It is not obvious why the RPL5 blot in figure 2b shows a very strong knockdown, but the knockdown shown in supplemental figure S3d is very modest. Western blots in figures 2b and S3d need molecular weight markers..
4. It is interesting that the authors detect high levels of p53 in untreated HeLa cells that sharply diminish after RPL5 knockdown (figure S3d). In HeLa cells, the p53 tumor suppressor pathway is abrogated due to the expression of E6 and E7 oncoproteins, which not only promote its degradation but also inhibit its transcriptional activity. For that reason, p53 in HeLa cells does not have an anti-proliferative effect seen in non-transformed cell lines. Therefore, the HeLa cell line may not be the best experimental system to draw conclusions about whether or not p53 plays a role in the proliferation decrease seen in RPL knockdowns.
5. It is counterintuitive that less Rpl5 would yield more ribosomes, as shown in the simulation. The simulation in figure 3e predicts a large accumulation of matured ribosomal subunits in the RPL5 depletion model (blue points). In reality, assembly of mature ribosomes requires RPL5 whose supply is diminished. This should be explained and could be tested experimentally if the authors believe this is a likely outcome.
6. Could the nucleolus have an accumulation of immature subunits that are not fully assembled and are not exported for that

reason, which leads to the expansion effect?

7. Figure S5 shows a simulation recapitulating the molecular dynamics in siRPL5 knockdown cells. It is not clear the dynamics of which marker is recapitulated in the simulation of rDNA particles.

8. DBA part of the study (Figures 5 and S6): DBA is characterized by red cell aplasia, but the WBC count (including lymphocytes) is usually normal. In other words, this is a bone marrow disorder manifested by the impaired erythropoiesis, not lymphopoiesis. Moreover, nucleoli in mature lymphocytes without signs of malignancy are small and not very prominent compared to the nucleoli in continuously proliferating HeLa cells that served as an experimental model for the rest of the study. Please explain the choice of lymphocytes, limitations and caveats of selecting these cells, and guide the reader how to think about this data relative to the bulk of the study in HeLa cells. It seems like comparing apples and oranges.

9. The text refers to "DBA patient with an RPL5 deletion". DBA is an autosomal dominant disorder in most cases, so this patient is likely heterozygous, i.e. suffers from haploinsufficiency, not a complete deletion.

10. How were lymphocytes identified for the analysis in Figure 5? Were blood smears also stained with the Wright's stain? The images in 5a appear less than great quality and could be larger to show more detail. Counter-stain for DNA or corresponding Wright's stain images could also be helpful to orient the reader. Also, lower right images in 5a (NPM1 in DBA) appears to be a colored image, while all others are in grayscale.

11. What are the area units (Y-axis) in figure 5b? Also, what is the reason that the NPM1 area in Control_1 is at zero? If the reason is that NPM1 stain could not be detected in this specimen, is it appropriate to include it in statistical analysis?

12. It is also not clear if data in 5c and 5d represent two patients or two specimens from the same patient (the same goes for controls). Were the controls age-matched?

13. Supplementary figure 6a: the Y axes denote the intensity (average or integrated?) of the nucleolus in DBA cells, which I assume is in raw arbitrary units, yet the legend reads "Quantification of subnucleolar areas". To accurately compare fluorescent intensities of specimens on different slides (if the labeling of the Y-axis in the figure is accurate), the signal should be normalized to something, because there could be a considerable variation in antibody labeling from slide to slide that is due to specimen processing, storage, etc. For intensity normalization, any lymphocyte antigen, or nuclear antigen (histone, for example) can be utilized.

14. In figures S6 and 5, the images do not advance the authors points. Furthermore, the sample size is not indicated. I am not confident that DBA lymphocyte data is sufficient in number and quality to support or refute the author's proposed mechanism of the role of RPL5 in the biophysical properties of the nucleolus. It is interesting to explore nucleolar properties in cells from DBA syndrome, but mature lymphocytes from the peripheral blood smear may not be the best model to investigate this.

15. The authors refer to the surface tension of the nucleolus throughout the manuscript but no direct measurements of this were made. This seems overclaimed, or they need to explain how they are inferring changes in surface tension (just from density??).

16. In Figure 6 the authors could use a lighter shade of green in the rpl5 knockdown cartoon to represent a lower density.

17. In Figure S4 the authors present staining of Cajal bodies and nuclear speckles, claiming these are normal with rpl5 knockdown. Is the claim based on gross visual inspection? Wndchrm? Please indicate how normality was assessed.

Reviewer #3 (Comments to the Authors (Required)):

In this report Matsumori et al. are addressing the role of ribosomal protein uL18 (RPL5) in nucleolar structural maintenance and ribosome biogenesis. They also describe how uL18 contributes to the biophysics of the nucleolus (mobility in the GC compartment), and they propose a role for the protein in rDNA compaction.

I have mixed feelings about this manuscript. On the one hand, half of the data was already published elsewhere (the part dealing with the role of uL18 in nucleolar structure and pre-rRNA processing - btw, the published work is insufficiently referred too), on the other, there are a few quality elements to the work (biophysics, TEM).

In modern eukaryotes, the nucleolus consists of three embedded layers. The most internal is called the fibrillar center (FC), it is surrounded by the dense fibrillar center (DFC); FC/DFC defining modules embedded in a third layer the granular component (GC). Pre-rRNA synthesis occurs at the interface between the FC and DFC. Thus, the rDNA is buried at the inner core of the organelle/condensate.

-One novel element of the work is that the mobility of NPM1, a GC component (the more peripheral compartment of the nucleolus), is increased upon uL18 depletion while that of FBL, a DFC component (the middle layer), is not (Fig 2d). The biophysics of the nucleolus in cells is shown to be similar to that of in vitro reconstituted structures (as judged by diffusion coefficient and diffusive exponent) - this is useful. The TEM analysis and its description is very nice (Fig 2a).

-Another novel element is that in reference cells, there are only a few rDNA foci (between 2-3) which are quite close (Figure 3a), while upon uL18 depletion the authors detect more rDNA foci which are also smaller and more dispersed throughout the nucleolar space. Hence the suggestion that uL18, a protein which enhances the mobility of a GC protein (NPM1) is involved in compacting the rDNA, which is located at the inner core of the nucleolus. But how this occurs remains unclear. There is quite a

distance between NPM1 in the GC at the rDNA at the inner core of the nucleolus, ...

In my opinion, to grant publication, the authors should provide direct experimental evidence that indeed rDNA compaction is affected in the absence of uL18. This would considerably strengthen their manuscript.

In addition, the authors should address the comments below.

General comment

-Similar published work should be better referenced:

Several medium/high throughput screens aimed at identifying factors important for nucleolar structure maintenance have been performed.

1) In one work, the eighty ribosomal proteins were systematically investigated for a role in nucleolar structure maintenance, mature rRNA accumulation, pre-rRNA processing, and p53 homeostasis. See DOI:10.1038/ncomms11390 and www.ribosomalproteins.com.

On this occasion an algorithm was specifically developed, the iNo scoring method, which offers unprecedented statistical power, unrivaled thus far for this type of analysis. Half of the conclusions of this submission are in this published work. Conclusions like 'uL18 is the most important ribosomal protein for nucleolar structure maintenance' or 'among proteins of the large ribosomal subunit those that assemble late are the most important ones for nucleolar structure' etc. were already spelled out in this published work, and should be put in perspective here for scholar balance.

2) In a second work the abundant nucleolar proteome (625 proteins) was tested for nucleolar structure; presumably there is a very large overlap between this set and the set of factors tested here. See DOI: 10.1038/s41596-018-0044-3. Again, it would be useful to compare the results (at least briefly).

3) Finally, in the iNo scoring method, a DFC marker (fibrillarin) was used and it was expressed from an additional locus. However, it was also shown in this work that a GC marker (PES1) detected with antibodies (endogenous protein) -similar to NPM1 used in this work- led to the same conclusions.

Specific comments:

-Figure 4: there is an inconsistency with regard the effects on rRNA synthesis.

The result in Figure 4 panel b (RTqPCR) shows there is an important reduction of the large precursor detected with the amplicons used (47S).

This is in contradiction with a previously published work (<https://www.ribosomalproteins.com/uL18/>) where it was shown that uL18(RPL5) depletion leads to strong accumulation of large precursor accumulation (45S/47S).

This is also in contradiction with the authors' own data. Indeed, Figure 4 panel e clearly shows that the 45S is accumulated upon RPL5 depletion. The authors should probe this gel in a northern blotting experiment to detect the 47S and 45S and they will confirm this. The accumulation of 32S, on the other hand, is consistent with former work.

-Figure 1:

In panel b, TIF1-A and POP4 are rooted together.

In panel a, TIF1-A depletion is very well-known to lead to formation of so-called "nucleolar caps" (these can reasonably be seen in panel a). Since POP4 appears so close on the clustering, should its depletion not also lead to cap formation (not visible on panel a)? May be higher magnification would help addressing this?

-There is a discussion on p53 homeostasis (Fig S3) which is inadequate since this work was performed in HeLa cells (which are not physiologically regulated for p53).

-In Figure 3, panel a, DAPI signal: is it just this cell or a more general observation that the DAPI signal is also affected upon uL18 depletion?

-Effects on nucleolar structure in DBA cells (Figure 5): I really appreciate the effort to describe the nucleolus in peripheral blood cells, which are more difficult to image. It is quite difficult to see nucleolar morphological alterations on the images. What is clearly apparent is a more intense labelling of the GC protein NPM1.

Re: Life Science Alliance manuscript #LSA-2021-01045-T

Point-by point responses are below. The reviewers' comments are italicized and highlighted in blue. For clarity, some comments are subdivided, and we responded to them individually.

Reviewer #1 (Comments to the Authors (Required)):

The authors conducted RNAi screen of 745 human genes that were previously associated with nucleolus, in search for genes involved in nucleolar structure. They identified 16 genes essential for normal distribution of nucleophosmin as proxy of nucleolar organisation. Image analysis with wndchrn was employed to classify the phenotype resulting from the RNAi of the 16 genes, not the 745 genes. The authors decided to focus on characterisation of RPL5, which led the authors conclude that RPL5 has roles in nucleolar organisation. A similar, but more extensive study was published by Nicolas et al. in 2015 (cited in the manuscript as ref 21), including roles of RPL5 and other ribosomal proteins in nucleolar integrity.

We appreciate the reviewer for this important comment.

>Image analysis with wndchrn was employed to classify the phenotype resulting from the RNAi of the 16 genes, not the 745 genes.

The reviewer is correct, and we clarified it and revised our abstract on page 2, lines 4-5.

> A similar, but more extensive study was published by Nicolas et al. in 2015 (cited in the manuscript as ref 21), including roles of RPL5 and other ribosomal proteins in nucleolar integrity.

The same critique was provided by reviewer #3, with detailed information and advice. To offer a perspective for our work and the previous findings, we now refer to the work by Dr. Lafontaine's group (Nicolas *et al.*, 2016 and Stamatopoulou *et al.*, 2018) in more detail, and describe the common and distinct points, in the introduction and discussion sections, on page 5, lines 3-11 and page 17, lines 6-21.

The previous work by Lafontaine's group employed the model-based (iNO scoring method) image analysis system that was specifically developed for nucleolar analyses. In our work, we used the model-free (wndchrn) system, without any steps specialized for the nucleolar morphology, including segmentation or additional algorithms or pipeline construction. Furthermore,

we screened proteins among 745 factors, in which 400 are in the nucleolar proteome and 345 are others that are involved in nuclear events and signaling pathways for gene regulation, energy metabolism and DNA repair (Boulon *et al.*, 2010), together with nucleolar components that had been previously identified by a proteome analysis (Andersen *et al.*, 2005). In contrast, the previous work screened 80 ribosomal proteins and 668 in the nucleolar proteome. In other words, this new survey evaluated approximately the same number of genes, but a broader set of gene types, with nearly half not being localized in the nucleolus. Additionally, the analysis employed in this study was not model-based, and could thus detect phenotypes other than those specified by the iNO scoring method.

To clarify the common and distinct findings, we have revised Table S1 by highlighting the proteins that were tested in both studies, in yellow. We found that five proteins were commonly identified to be important for nucleolar morphologies, and 11 proteins were identified only in our work. We have added the following sentences on page 17, line 22-page 18, line 3 in the revised text.

“In another study, the same group screened factors that are involved in nucleolar structure maintenance among 668 abundant nucleolar proteins, using the iNo scoring method, and identified 86 proteins as being important (Stamatopoulou *et al.*, 2018). Among them, 52 were also in our siRNA library (highlighted in yellow in **Table S1**). Five of them (RPL5, RPL27A, NPM1, WDR43 and TIF-IA) were among the 16 proteins that we found to be important for nucleolar morphology (**Fig. 1a-b**), again solidifying the importance of RPL5 in the nucleolus.”

Regardless of the fundamentally different image analysis methods and few overlapped factors identified, RPL5 was commonly found to be an important factor for the nucleolus. These results solidify the significance of RPL5. They also suggest the versatility of the wndchrm image system in applications for other structures in the cell. We hope that the reviewer could kindly consider that these are novelties of our work and provide relevant information for researchers.

Most importantly, as reviewer #3 mentioned in a fair manner, our work includes novel elements, including the transmission electron microscopy (TEM) analysis (**Fig. 2a**) and the nucleolar morphology in the DBA patient with the heterozygous RPL5 deletion (**Fig. 5**). We also showed that the mobility of NPM1, a GC component, is increased upon the RPL5 depletion while that of FBL, a DFC component, is not (**Fig. 2d**). Our work also demonstrated that the biophysical features of the nucleolus in cells are similar to those of *in vitro* reconstituted structures (as judged by diffusion coefficient and diffusive exponent). These are novel findings that were not reported in the previously published works.

Specific points

(1) The initial RNAi screen is not conducted carefully. The depletion of mRNA for each RNAi should be tested in order to evaluate the phenotype.

According to this comment, we added the qRT-PCR data in the new **Fig. S1c**. We also added or modified the sentences explaining how we controlled the screening in the Results section, or Materials and methods section, as follow.

On page 6, lines 14-18: “To secure each knockdown, we used a method with pooled siRNAs including four unique siRNAs designed for one target gene, which guarantees a targeted knockdown. To monitor the appropriate RNAi knockdowns, each set of experiments always contained a pair of negative (siGL3, non-targeting sequence) and positive (siLMNA) controls (see **Materials and methods** for details).”

On page 20, lines 8-19: “To evaluate the RNAi knockdown efficiency, each 96-well plate contained a well for a non-targeting siRNA with the sequence CUUACGCUGAGUACUUCGA (GL3, Nippon EGT) as the negative control. The plate also contained a well for the positive control, siLMNA (Dharmacon siGENOME SMARTpool: M-004978-01-0005), which was immunostained with anti-LaminA/C under the same conditions used for the other siRNAs.”

In summary, we have conducted our initial siRNA screen with careful considerations.

(2) p6 lines 6-7

Fig S1b shows data on intensity and area of NPM1 signals - not morphology.

We changed the word “morphology” to “intensity and area of NPM1 signals” in the corresponding sentence on page 6, lines 19-20.

(3) Fig S1b

What are the data distribution? The 16 selected genes do not appear to show any particular features in these data set. It seems that the selection criteria is independent of quantification in Fig S1b.

> (3)-1: What are the data distribution?

We apologize for our insufficient description. First, to respond to this critique as well as the point mentioned in comment (5), we re-aligned our quantification data in an alphabetical order in **Table S1**. In this table, we simply described how the nucleolar intensity and area changed upon each knockdown.

> (3)-2: The 16 selected genes do not appear to show any particular features in these data set. It seems that the selection criteria is independent of quantification in Fig S1b.

We chose them because their knockdowns reproducibly changed nucleolar morphologies, based on eye-inspection. We now clearly describe this on page 7, lines 2-3, as follows.

“Among the examined factors, we found that the individual losses of 16 proteins reproducibly changed the nucleolar intensities and shapes, based on eye-inspections..”

(4) p6 line 7-8

What are the true criteria for selection of the 16 genes?

We chose the 16 genes because the morphologies were clearly and reproducibly changed upon knockdown, as we described above in our response to comment # (3)-2. Their nucleoli showed combinations of changes: convexed-hulled (siRPL28, siRPL5, siRPL27A, siRPL7, siRPL21, siRPL6, and siPLD2, and siWBP11), rounded-up (siTIF-1A, siPOP4, siELK1, and siTRM28), or their signals were reduced (siNPM1, siTYMP, siRHOC, and siWDR43), as shown below. A more thorough quantification was performed with the wndchrm algorithm, as shown in **Fig. 1b** and **c**.

[Figure removed by editorial staff per authors' request].

(5) Table S1 and Fig S1b

It is more useful if the data is sorted by rank or in alphabetical order, but not the index number. It was difficult to find genes of my interest e.g. RPL11 or NCL from the list.

According to this comment, we sorted the data in alphabetical order in **Table S1** and **Fig. S1b**.

(6) p7 line 2 / Fig S3a

Are their mRNA expression reduced as much as siRPL5?

We performed qRT-PCR analyses, and the results are shown in the new **Fig. S3b**. The mRNA expressions of siRPL9, siRPL13, and siRPS9 were reduced as much as that of RPL5. We describe this result on page 7, lines 21-23.

(7) Fig S3b

Please show the area of nucleolus.

We understand that it is important to simultaneously visualize RPL5 and the nucleolar marker. Therefore, we tried an immunofluorescence analysis with anti-RPL5 and NPM1 antibodies. However, the currently available lot of anti-RPL5 rabbit polyclonal antibodies (GeneTex, GTX10821) did not work reproducibly. Because we could not perform additional relevant experiments, we removed this figure and the corresponding sentence, “An immunofluorescence analysis demonstrated that a portion of RPL5 is in the nucleolus (**Fig. S3b**), reflecting its transient nucleolar localization before it is exported to the cytoplasm and engaged in protein translation.” on page 7, lines 16-18 in the previous manuscript. Please note that this revision does not change the main conclusion of our paper.

(8) Fig 1c

What are the morphological distances for other RPL, e.g. RPL9 and RPL13 ?

We did not capture enough photos for other RPLs, and could not measure their morphological distances by the wndchrn analysis.

(9) P6 lines 17-21

Which parameters were selected to define groups and draw the dendrogram ?

We have created the new **Table S2**, showing a representative set of 431 image features automatically extracted and weighted for analysis for the dendrogram in **Fig. 1b**. We also described it in the text on page 7, line 12.

(10) Discrepancy between Fig S3g and Fig 1b

siRPL5 and siRPL21 look similar in Fig3g, but they are mapped on different branches and siControl is mapped in between in Fig 1b. Are the parameters to calculate the distance same in Fig S3g and Fig 1b?

We appreciate this insightful comment. The parameters to calculate each image in **Fig. S3g** and **Fig. 1b** were the same, 2,873 feature values. However, the image features automatically extracted and weighted for the best classification based on the Fisher discrimination scores for the current **Fig. 3h** (previous **Fig. S3g**) and **Fig. 1b** were different, due to different sets of images analyzed. This may be the reason why we observed that *siControl* is more similar to CHX, DMSO, and *siControl/DMSO* than *siRPL5*, which is more similar to *siRPL21*. Please note that these differences do not change any of our conclusions.

This reviewer's comment prompted us to create the new **Table S3**, showing a representative set of 431 image features automatically extracted and weighted for analysis in **Fig. 3h** (previous **Fig. S3g**), which can be compared with **Table S2**. We now describe it on page 9, line 5.

(11) Fig 2a is not compelling evidence for RPL5 as a LLPS facilitator.

We agree with the reviewer that **Fig. 2a** itself is not compelling evidence for RPL5 as an LLPS facilitator. We toned down the sentence on page 10, lines 3-5 in the previous manuscript, and cited another *in vitro* analysis to support our claim, as follows.

“The lower molecular density, the loss of sphericity, and the less distinct interface of the GC region suggested that its biophysical nature might be altered in the absence of RPL5, as shown *in vitro* (Mitrea *et al.*, 2016; Mitrea *et al.*, 2014).”

We also revised the following sentence “Single-molecule tracking analyses and EM observations demonstrated a novel contribution of RPL5 to a biophysical property of the GC region in the nucleolus of the somatic cell”. We removed “and EM observations” from it. The revised sentence is currently on page 11, lines 5-6.

(12) Fig S3d The intensity for histone H3 is not shown.

We quantified and added the intensity for histone H3 in the new **Fig. S3c** (previous **Fig. S3d**).

(13) Fig S3e Growth curves are easier to interpret if drawn in log scale.

According to this comment, we re-drew the growth curves in log scale (new **Fig. S3f**, corresponding to the previous **Fig. S3e**). We also included a new growth curve in the new **Fig. S3e**, according to the comment from reviewer #2. We have also drawn it in log scale, according to this reviewer's request. We appreciate the important advice.

(14) Fig S4

siControl cells passing through mitosis are more appropriate control to show that aberrant nucleoli are phenotype attributed to siRPL5, not a cell cycle event.

We appreciate the reviewer for this important comment. Accordingly, we have performed new live-cell imaging experiments, using *siControl* and *siRPL5* cells that passed through mitosis. We agree that this current experiment is more appropriate to show that aberrant nucleoli are a phenotype attributed to *siRPL5*, rather than a cell cycle event. We created the new **Fig. S4a and b**, and replaced the previous one. We now show that a control cell contains multiple nucleoli both before and after mitosis (new **Fig. S4a**). In the *RPL5*-depleted cells, the nucleolus was integrated into one aberrant, large structure, only after the cells passed through mitosis (new **Fig. S4b**). We rewrote the figure legends for the new **Fig. S4a and b**.

(15) I am surprised with notable absence of reference to RPL11 - a well known cognate subunit of RPL5.

We apologize that we did not include the siRNA for RPL11 in our library. Therefore, we do not have a result for RPL11. We have added the following sentence in the discussion section, on page 19, lines 3-6.

“It may be consistent that the lack of the cognate subunit of RPL5, RPL11, showed a similar effect on nucleolar morphology (Nicolas *et al.*, 2016). RPL5 and RPL11 are functionally related and physically interact in the pre-ribosomal complex (Donati *et al.*, 2013; Kressler *et al.*, 2012).”

Reviewer #2 (Comments to the Authors (Required)):

In this work, the authors used siRNA screening combined with machine learning to find genes whose knockdown affects the morphology of the nucleolus. The screen was conducted in HeLa cells using nucleolar protein NPM1 as a readout. Data were classified with an open-source software Wndchrm. Hits included a group of RPL proteins that are integral components of the ribosome. Their knockdowns shared common morphological features, most notably enlarged size and loss of sphericity of the nucleolus.

Authors attribute RPL5 knockdown to imperfect phase separation, speculating that RPL5 may contribute to the elasticity and surface tension of the Granular Component (GC) of the nucleolus, and can influence its viscosity that was measured by single-molecule tracking of SNAP-tagged NPM1. Authors propose that RPL5 facilitates interactions between rDNA, itself, and NPM1, thereby promoting liquid-liquid phase separation and expression of rRNA. In the end, the authors explored nucleolar features in lymphocytes from a patient(s) suffering from Diamond Blackfan anemia (DBA).

Many experiments in this paper appear to be conclusive and well-done, such as electron microscopy and single-particle tracking analysis.

We greatly appreciate this constructive comment.

In my opinion LLPS is still at the hypothesis stage for nucleoli, and not a foregone conclusion for how they organize. I don't see why the observed features of several RPL knockdowns must be attributed specifically to LLPS and cannot be caused simply by abnormal accumulation of partially assembled ribosomal subunits that are not fully assembled due to lack of key subunits, and are retained in the nucleolus. The granular component is the compartment where the last stages of assembly of pre-ribosomal particles occur, preceding their transport out of the nucleolus. It is plausible that in the absence of given RPLs this enlarged GC with decreased density may be filled with more unassembled or partially assembled ribosomal subunits and fewer assembled pre-ribosomal particles. Abnormal accumulation of unassembled proteins can also impair overall nucleolar function, including transcription. To further test the involvement of RPLs in LLPS the authors could include in vitro experiments with isolated components, but I think that is outside the scope of the current manuscript. Previous in vitro studies with NPM1 showed that it can undergo homotypic and heterotypic LLPS at sufficient concentrations in systems that did not contain RPLs. Also, more studies are needed to show that LLPS directly facilitates rDNA transcription by PolI in vivo. My suggestion to authors is to be wary of interpreting current results solely from the

perspective of LLPS. At a minimum they should provide an alternative explanation of their findings that does not invoke LLPS.

We agree with the reviewer that we have to be careful when interpreting our current results solely from the perspective of LLPS, and provide an alternative explanation of our findings. Accordingly, we have added new sentences on page 19, lines 10-14, as below.

“Another possibility is that the features of the RPL5 knockdown are caused by the abnormal accumulation of partially assembled ribosomal subunits that are retained in the nucleolus. The enlarged GCs with decreased particle-density may be due to the increased presence of partially assembled ribosomal subunits that interact less with each other. Either way, proper nucleolar properties are necessary for rDNA bundling, rRNA transcription and processing.”

It would be significantly strengthen the authors findings to confirm the RPL knockdown phenotype in cell lines other than HeLa. HeLa cells are highly aneuploid and adapted to constant protein dosage imbalance, and their p53 pathway is not functional. Besides, nucleolar size and shape in different cell lines can vary. It would significantly strengthen the manuscript to confirm the RP knockdowns in HeLa cells, and confirm/extend the knockdowns and nucleolar phenotype in a minimum of one additional cell line that is karyotypically normal.

We appreciate the reviewer for this comment, and agree that data obtained with a different cell type would strengthen our manuscript. Accordingly, we performed several new experiments using RPE-1 cells, a human retinal pigment epithelium cell line that is karyotypically normal and expresses wild type p53. We monitored the level of p53 (new **Fig. S3c**), and confirmed the nucleolar phenotype (new **Fig. S3g**) and the cell proliferation (new **Fig. S3e**), all under the RPL5 knockdown conditions. We described the results in the manuscript on page 8, lines 11-23.

Other points:

(1) The quality of EM images in figure 2a is impressive and provides strong evidence for decreased GC density in RPL5 knockdown. This can be interpreted as a decreased density of mature ribosomal subunits that appear as small dark granules on TEM labeled with uranyl acetate. I do not see the triangles referred to in the figure legend.

>(1)-1: The quality of EM images in figure 2a is impressive and provides strong evidence for decreased GC density in RPL5 knockdown.

We greatly appreciate the very positive comment.

>(1)-2: This can be interpreted as a decreased density of mature ribosomal subunits that appear as small dark granules on TEM labeled with uranyl acetate.

We agree with the reviewer. We have added a new sentence describing this interpretation, on page 9, lines 21-23, as follows.

“This may be due to the decreased density of mature ribosomal subunits, which appear as small dark granules on TEM labeled with uranyl acetate.”

>(1)-3: I do not see the triangles referred to in the figure legend.

We lost the triangles during the course of the first submission; therefore, we added them back in **Fig. 2a**. Thank you for pointing this out.

(2) Only Rpl5 knockdown is confirmed by Western. Some journals require that phenotypes are observed with at least two unique siRNAs to avoid reporting off-target effects. In this study a smart pool was used with 4 separate siRNAs. The authors should consider confirming their hits (at least RP) with individual siRNAs and corresponding Western blotting.

According to this comment, we have performed new experiments using cells treated with individual RNAs rather than the smart pool, siRPL5 #1 and #2, and siRPL21 #1 and #2 (new **Fig. S1c and d**). By western blotting and qRT-qPCR, we confirmed that the protein and mRNA levels were decreased with each individual siRNA. By immunofluorescence, we then confirmed that the nucleolar phenotypes were the same with the individual siRNAs and the smart pool. All of these results are shown in the new **Fig. S1c and d** and described in the text, on page 7, lines 5-7.

(3) It is not obvious why the RPL5 blot in figure 2b shows a very strong knockdown, but the knockdown shown in supplemental figure S3d is very modest. Western blots in figures 2b and S3d need molecular weight markers.

> (3) -1: It is not obvious why the RPL5 blot in figure 2b shows a very strong knockdown, but the knockdown shown in supplemental figure S3d is very modest.

We agree with the reviewer that the knockdown efficiency was inconsistent in **Fig. 2b** and the previous **Fig. S3d**. Because we performed more experiments to revise the previous **Fig. S3d** (current **Fig. S3c**) to address another comment provided by this and another reviewer (reviewer #3), we did so more carefully. Now, we have comparable knockdown efficiencies: 0.1-fold reduction in **Fig. 2b** and 0.2-fold reduction in **Fig. S3c** (previous **Fig. S3d**). Our claim does not change with this revision.

> (3) -2: Western blots in figures 2b and S3d need molecular weight markers.

We added molecular weight markers for western blots in **Fig. 2b** and the new **Fig. S3c** (previous **Fig. S3d**). We thank this reviewer for their careful inspection of our manuscript.

(4) It is interesting that the authors detect high levels of p53 in untreated HeLa cells that sharply diminish after RPL5 knockdown (figure S3d). In HeLa cells, the p53 tumor suppressor pathway is abrogated due to the expression of E6 and E7 oncoproteins, which not only promote its degradation but also inhibit its transcriptional activity. For that reason, p53 in HeLa cells does not have an anti-proliferative effect seen in non-transformed cell lines. Therefore, the HeLa cell line may not be the best experimental system to draw conclusions about whether or not p53 plays a role in the proliferation decrease seen in RPL knockdowns.

We appreciate this comment. We understand that the HeLa cell line is not an appropriate system, as this and another reviewer (reviewer #3) pointed out. Therefore, we performed the same experiment with diploid RPE-1 cells, which were originally derived from human retinal pigment epithelial cells and have the normal p53 signaling pathway. The data are now shown in the new **Fig. S3c-e**. In RPE-1 cells, p53 protein accumulation was accompanied with p21 accumulation, upon the depletion of TIF-IA due to nucleolar stress (**Fig. S3c, right**). This confirms that the p53 signaling pathway is functional in this karyotypically normal cell, even though the p53 protein did not remarkably accumulate upon the RPL5 depletion in RPE-1 cells. We observed the aberrant nucleolar morphologies in the RPL5-depleted RPE-1 cells (new **Fig. S3d**). Cell growth was also inhibited with the RPL5 knockdown in both the HeLa and RPE-1 cell lines (new **Fig. S3e and S3f**). These results suggest that the cell growth inhibition and aberrant nucleolar morphology upon the RPL5 knockdown are independent of the p53 signaling pathway. These findings are now described in the text, on page 8, lines 11-23. We have also created new figure legends for **Fig. S3c-e**.

(5) It is counterintuitive that less Rpl5 would yield more ribosomes, as shown in the simulation. The simulation in figure 3e predicts a large accumulation of matured ribosomal subunits in the RPL5 depletion model (blue points). In reality, assembly of mature ribosomes requires RPL5 whose supply

is diminished. This should be explained and could be tested experimentally if the authors believe this is a likely outcome.

We thank the reviewer for raising this important issue. With this comment, we realized that the label “matured ribosome” in the siRPL5 model in **Fig. 3d-e** is inappropriate, because we actually mean distributions of virtual crowders, or premature ribosome-like crowders, which must be considered in the calculations, as follows.

In this manuscript, we constructed and simulated an siRPL5 model to reveal the contributions of specific properties of RPL5 to regulate the nucleolus organization and intra-nucleolus molecular motions. We assumed that the siRPL5 model contained crowders, instead of matured ribosomes, to make the size of the simulation space and the total volume fraction of particles in the siRPL5 the same as in the control model. This could be partially assembled ribosomal subunits that are retained in the nucleolus.

Additionally, we assumed that each crowder has the same physical properties as a matured ribosome but could not be replaced with RPL5 as in the control model. By these assumptions, we could evaluate the pure effects of RPL5 on the dynamic features of the nucleolus through the comparison between the control and siRPL5 models.

Thus, we replaced “matured ribosome” with “crowder” in **Fig. 3d-e** in the revised manuscript. We also revised the Materials and methods section on page 31, lines 9-12 and lines 16-18, and the legend for **Fig. 3d-e** on page 41, lines 18-19.

(6) Could the nucleolus have an accumulation of immature subunits that are not fully assembled and are not exported for that reason, which leads to the expansion effect?

We agree with this comment. As mentioned above, we added this important possibility in the text, on page 19, lines 10-14, as follows.

“Another possibility is that the features of the RPL5 knockdown are caused by the abnormal accumulation of partially assembled ribosomal subunits that are retained in the nucleolus. The enlarged GCs with decreased particle-density may be due to the increased presence of partially assembled ribosomal subunits that interact less with each other. Either way, proper nucleolar properties are necessary for rDNA bundling, rRNA transcription and processing.”

(7) Figure S5 shows a simulation recapitulating the molecular dynamics in siRPL5 knockdown cells. It is not clear the dynamics of which marker is recapitulated in the simulation of rDNA particles.

We think that the dynamics of fibrillarin (FBL) in the cells shown in **Fig. 2** are recapitulated by the simulation of rDNA particles in **Fig. S5**. This is because FBL is part of the dense fibrillar component (DFC) that resides in the interior of the nucleolus, and rDNA is transcribed on the border of FC/DFC (Yao et al., 2017). For clarification, we added the following sentence in the revised text, on page 13, lines 10-12.

“Therefore, the dynamics of NPM1 in cells were well recapitulated with the NPM1 particles, and those of FBL in cells were recapitulated by the rDNA particles in our simulation. ”

(8) DBA part of the study (Figures 5 and S6): DBA is characterized by red cell aplasia, but the WBC count (including lymphocytes) is usually normal. In other words, this is a bone marrow disorder manifested by the impaired erythropoiesis, not lymphopoiesis. Moreover, nucleoli in mature lymphocytes without signs of malignancy are small and not very prominent compared to the nucleoli in continuously proliferating HeLa cells that served as an experimental model for the rest of the study. Please explain the choice of lymphocytes, limitations and caveats of selecting these cells, and guide the reader how to think about this data relative to the bulk of the study in HeLa cells. It seems like comparing apples and oranges.

We agree with this reviewer, and consider this comment to be very important. Although analyzing hematopoietic cells from bone marrow may be ideal, it is not a realistic endeavor. However, the human leukemic K562 cell line may offer a good model system, because these cells can undergo erythroid differentiation *in vitro*, and thus it serves a model system for erythroid differentiation in which DBA has a defect (Luo *et al.*, 2017). As a matter of fact, we had already knocked down RPL5 in K562 cells, and analyzed their nucleoli by immunofluorescence with anti-NPM1 antibodies. We found that the nucleolar GC region becomes larger upon the RPL5 depletion, as seen in the patient lymphocytes and HeLa cells. We have added these data in the new **Fig. S6c**, and describe them on page 15, lines 10-15. We also revised the legend for **Fig. S6c**.

The loss-of-function mutation in RPL5 is frequent found in patients with the congenital defect (Kampen et al., 2020). This suggests that the nucleolar aberrancy could be transmitted to the whole body. In fact, observations of other cell types, including lymphocytes derived from DBA patients, offer useful knowledge (Gazda *et al.*, 2008). We now describe these findings in the text on page 15, lines 15-17. We also revised the legend for **Fig. S6c**.

(9) The text refers to "DBA patient with an RPL5 deletion". DBA is an autosomal dominant disorder in most cases, so this patient is likely heterozygous, i.e. suffers from haploinsufficiency, not a complete deletion.

The reviewer is correct. Therefore, we revised the Abstract and Results sections, as follows.

On page 2, line 12: “a heterozygous,”

On page 19, line 17: “DBA patient with a heterozygous RPL5 deletion”

(10) How were lymphocytes identified for the analysis in Figure 5? Were blood smears also stained with the Wright's stain? The images in 5a appear less than great quality and could be larger to show more detail. Counter-stain for DNA or corresponding Wright's stain images could also be helpful to orient the reader. Also, lower right images in 5a (NPM1 in DBA) appears to be a colored image, while all others are in grayscale.

>(10)-1: How were lymphocytes identified for the analysis in Figure 5?

We identified lymphocytes in the specimen by Giemsa staining, as shown in the figure below.

[Figure removed by editorial staff per authors' request].

>(10)-2: The images in 5a appear less than great quality and could be larger to show more detail. Counter-stain for DNA or corresponding Wright's stain images could also be helpful to orient the reader.

According to the comment, we have enlarged the images in **Fig. 5a** to show more detail. We also revised **Fig. 5a** by replacing it with the merged images of NPM1 and the counter-stain for DNA, to orient the reader.

>(10)-3: Also, lower right images in 5a (NPM1 in DBA) appears to be a colored image, while all others are in grayscale.

We sincerely thank this reviewer for their careful inspection of our manuscript. We replaced the lower right images in **Fig. 5a** (NPM1 in DBA), and now show all the data in color.

(11) What are the area units (Y-axis) in figure 5b? Also, what is the reason that the NPM1 area in Control_1 is at zero? If the reason is that NPM1 stain could not be detected in this specimen, is it appropriate to include it in statistical analysis?

The area unit (Y-axis) in **Fig. 5b** is number of pixels. To clarify it, we revised the labeling in **Fig. 5b** to “Area of the nucleolus (number of pixels)”.

The reason why the NPM1 area in Control_1 is at zero is because the staining signal was weak in most of the cells tested. Among 21 cells tested, the median value of Control 1 was 0, and the maximum value was 474. The median value of Control 2 was 86, and the maximum value was 1192. Please note that all of the immunofluorescences for Control 1, Control 2 and DBA in **Fig. 5a** were processed side-by-side, at the same time on the same date, under identical conditions, including the antibody and buffer, microscopic set-up, photo-capturing and image analysis set-ups, and so on. We think that it is meaningful that the value of Control 1 for GC was low.

(12) It is also not clear if data in 5c and 5d represent two patients or two specimens from the same patient (the same goes for controls). Were the controls age-matched?

> (12)-1: It is also not clear if data in 5c and 5d represent two patients or two specimens from the same patient (the same goes for controls).

In **Fig. 5c and d**, data from one patient (DBA) and two normal individuals (Control 1 and Control 2) are shown. For each individual, 40 images of NPM1 were collected, and they were divided into two-subgroups: DBA_1 and _2, Control 1_1 and _2, and Control 2_1 and _2.

We admit that this was not clear in the previous manuscript, because the labels for **Fig. 5** were inappropriate. We fixed them in the revised version. For example, we replaced “Control_1” in **Fig. 5b** with “Control 1”. In addition, we revised the figure legend for **Fig. 5**, on page 42, lines 19-20.

> (12)-2: Were the controls age-matched?

The controls are not age-matched with the patient. The DBA patient was an infant, and the age-matched control samples were not available.

(13) Supplementary figure 6a: the Y axes denote the intensity (average or integrated?) of the nucleolus in DBA cells, which I assume is in raw arbitrary units, yet the legend reads "Quantification of subnucleolar areas". To accurately compare fluorescent intensities of specimens on different slides (if the labeling of the Y-axis in the figure is accurate), the signal should be normalized to something, because there could be a considerable variation in antibody labeling from slide to slide that is due to specimen processing, storage, etc. For intensity normalization, any lymphocyte antigen, or nuclear antigen (histone, for example) can be utilized.

> (13)-1: Supplementary figure 6a: the Y axes denote the intensity (average or integrated?) of the nucleolus in DBA cells, which I assume is in raw arbitrary units, yet the legend reads "Quantification of subnucleolar areas".

The reviewer's assumption is correct. The Y axes denote the average intensity of the nucleolus in arbitrary units. Therefore, we revised the legend for **Fig. S6a**, as follows.

“Signal intensities of the subnucleolar markers in the DBA patient and healthy control individuals, in arbitrary units (a.u.)”

> (13)-2: To accurately compare fluorescent intensities of specimens on different slides (if the labeling of the Y-axis in the figure is accurate), the signal should be normalized to something, because there could be a considerable variation in antibody labeling from slide to slide that is due to specimen processing, storage, etc. For intensity normalization, any lymphocyte antigen, or nuclear antigen (histone, for example) can be utilized.

We completely agree with the reviewer. It is best to normalize the signals to accurately compare the fluorescent intensities of specimens on different slides. However, this was technically difficult, because we could not find the appropriate control antibodies that work with UBF, FBL, and NPM1, under the same immunofluorescence conditions. Instead, as the second best approach, we very carefully performed immunofluorescence analyses for the controls and the patient in a side-by-side manner on the same day, under exactly the same conditions. Images were also captured on the same day under identical microscopic conditions.

(14) In figures S6 and 5, the images do not advance the authors points. Furthermore, the sample size is not indicated. I am not confident that DBA lymphocyte data is sufficient in number and quality to

support or refute the author's proposed mechanism of the role of RPL5 in the biophysical properties of the nucleolus. It is interesting to explore nucleolar properties in cells from DBA syndrome, but mature lymphocytes from the peripheral blood smear may not be the best model to investigate this.

> (14)-1: In figures S6 and 5, the images do not advance the authors points. Furthermore, the sample size is not indicated.

We now describe the sample size. For **Fig. 5b**, n=20, and for **Fig. 5c** and **d**, n=40. For **Fig. S6a**, n=20, and for **Fig. S6b**, n=40. All are now described in the corresponding figure legends.

> (14)-2: I am not confident that DBA lymphocyte data is sufficient in number and quality to support or refute the author's proposed mechanism of the role of RPL5 in the biophysical properties of the nucleolus. It is interesting to explore nucleolar properties in cells from DBA syndrome, but mature lymphocytes from the peripheral blood smear may not be the best model to investigate this.

We greatly appreciate that the reviewer finds it interesting to explore the nucleolar properties in cells from a DBA syndrome patient, which we intended in this work.

At the same time, we realized that we should tone down our interpretation, because it is not our intention to propose a role of RPL5 in the biophysical properties of the nucleolus, directly from our DBA lymphocyte data. With this reviewer's comment, we noticed that it was misleading to describe our conclusion, right after the DBA analysis section on page 13, lines 19-22 of the previous manuscript. Therefore, we relocated the following sentence to the discussion section on page 16, lines 9-12 in the revised manuscript.

“Our study revealed that specific RPLs, which have long been regarded as simple transient residents of the nucleolus, are actually critically involved in nucleolar assembly and function. Our data suggested that RPL5 functions in spatially positioning and condensing the chromosome sites for rDNA arrays in the nucleolus, and facilitates rRNA transcription and processing.”

> (14)-3: It is interesting to explore nucleolar properties in cells from DBA syndrome

We greatly appreciate this encouraging comment.

> (14)-4: but mature lymphocytes from the peripheral blood smear may not be the best model to investigate this.

We analyzed human leukemic K562 cells, as we mentioned for comment #8 by this reviewer. K562 cells are able to undergo erythroid differentiation *in vitro*, and thus offer a model system for erythroid differentiation in which DBA has a defect (Luo et al., 2017). We knocked down RPL5 in K562 cells, and analyzed their nucleoli by immunofluorescence with anti-NPM1 antibodies (new **Fig. S3c**). We found that the nucleolar GC region becomes larger upon the RPL5 depletion, as seen in the patient lymphocytes and HeLa cells. Loss-of-function mutations in RPL5 are frequent in patients with the congenital defect (Kampen et al., 2020), suggesting that the nucleolar aberrancy could be transmitted to the whole body. Observations of other cell types, including lymphocytes, offer useful knowledge. We now describe these points in the text on page 15, lines 10-17. We also revised the legend for **Fig. S6c**.

With this important comment, we think that it would be best to analyze human hematopoietic stem cells that are undergoing erythropoiesis differentiation. However, based on the difficulty in obtaining and handling those cells, this is currently beyond the scope of this study.

(15) The authors refer to the surface tension of the nucleolus throughout the manuscript but no direct measurements of this were made. This seems overclaimed, or they need to explain how they are inferring changes in surface tension (just from density??).

We agree with the reviewer. Previously, we mentioned surface tension on page 7 (line 13), page 16 (line 8), and page 37 (line 7), in the previous manuscript. We removed all of these from the revised text.

(16) In Figure 6 the authors could use a lighter shade of green in the rpl5 knockdown cartoon to represent a lower density.

According to this comment, we changed the model cartoon in **Fig. 6**.

(17) In Figure S4 the authors present staining of Cajal bodies and nuclear speckles, claiming these are normal with rpl5 knockdown. Is the claim based on gross visual inspection? Wndchrm? Please indicate how normality was assessed.

We claimed it based on eye-inspection. We now mention it on page 9, line 10 in the revised text.

Reviewer #3 (Comments to the Authors (Required)):

In this report Matsumori et al. are addressing the role of ribosomal protein uL18 (RPL5) in nucleolar structural maintenance and ribosome biogenesis. They also describe how uL18 contributes to the biophysics of the nucleolus (mobility in the GC compartment), and they propose a role for the protein in rDNA compaction.

I have mixed feelings about this manuscript. On the one hand, half of the data was already published elsewhere (the part dealing with the role of uL18 in nucleolar structure and pre-rRNA processing - btw, the published work is insufficiently referred too), on the other, there are a few quality elements to the work (biophysics, TEM).

> On the one hand, half of the data was already published elsewhere (the part dealing with the role of uL18 in nucleolar structure and pre-rRNA processing - btw, the published work is insufficiently referred too),

We thank this reviewer for providing valuable comments and important knowledge, which made us carefully revise and improve our manuscript.

Prior to responding to each comment, we would like to politely ask the reviewer to allow us to use the term “RPL5” to refer to uL18, for the sake of consistency with the text in our manuscript.

We apologize for our insufficient references to the previous reports. In our responses to this reviewer’s comments, we revised the text intensively and referred to the previous important works sufficiently. We have addressed and clarified the common and different findings. We clarified that we and others used fundamentally different image analysis methods, and identified few overlapped factors; however, both found RPL5 to be important. We consider this to be relevant information for the research field. We appreciate this reviewer for providing the detailed information on the previous publications.

> on the other, there are a few quality elements to the work (biophysics, TEM).

We greatly appreciate this constructive comment.

In modern eukaryotes, the nucleolus consists of three embedded layers. The most internal is called

the fibrillar center (FC), it is surrounded by the dense fibrillar center (DFC); FC/DFC defining modules embedded in a third layer the granular component (GC). Pre-rRNA synthesis occurs at the interface between the FC and DFC. Thus, the rDNA is buried at the inner core of the organelle/condensate.

-One novel element of the work is that the mobility of NPM1, a GC component (the more peripheral compartment of the nucleolus), is increased upon uL18 depletion while that of FBL, a DFC component (the middle layer), is not (Fig 2d). The biophysics of the nucleolus in cells is shown to be similar to that of in vitro reconstituted structures (as judged by diffusion coefficient and diffusive exponent) - this is useful. The TEM analysis and its description is very nice (Fig 2a).

-Another novel element is that in reference cells, there are only a few rDNA foci (between 2-3) which are quite close (Figure 3a), while upon uL18 depletion the authors detect more rDNA foci which are also smaller and more dispersed throughout the nucleolar space.

We greatly appreciate the very constructive comment. We are glad that the reviewer found the TEM and biophysics data to be useful, and our finding that rDNA foci are smaller and more dispersed upon RPL5 depletion to be a novel element.

Hence the suggestion that uL18, a protein which enhances the mobility of a GC protein (NPM1) is involved in compacting the rDNA, which is located at the inner core of the nucleolus. But how this occurs remains unclear. There is quite a distance between NPM1 in the GC at the rDNA at the inner core of the nucleolus, ...

We are in fact confused by the comment, “uL18, a protein which enhances the mobility of a GC protein (NPM1)...”, because our conclusion is the opposite. Our single molecule tracking analysis in **Fig. 2** showed that the knockdown of RPL5 enhanced the mobility of NPM1. This effect was further recapitulated by the computational simulation in **Fig. S5**. Therefore, we claim that RPL5 represses the mobility of a GC protein (NPM1), and confers anomalous diffusion in control cells. To clarify this, we added the following sentence on page 11, lines 4-5.

“Therefore, RPL5 represses the NPM1 mobility to confer anomalous diffusion in control cells. ”

With this fact, we hope that the reviewer understands our interpretation that RPL5 contributes to condensing GC. Because rDNA is located at FC/DFC, which is embedded inside of GC, rDNAs can be more condensed/compacted by the condensed outside environment/shell, GC. Conversely, rDNAs on FC/DFC will be de-condensed upon RPL5 depletion, because of the more relaxed environment/shell.

In my opinion, to grant publication, the authors should provide direct experimental evidence that indeed rDNA compaction is affected in the absence of uL18. This would considerably strengthen their manuscript.

To provide experimental evidence that rDNA compaction is indeed affected in the absence of RPL5, we first added more images of immuno-FISH analyses of rDNA and NPM1 in the control (siControl) and the RPL5-depleted cells (siRPL5) in the new **Fig. S4d**. They now clearly show that the rDNA array is more de-compacted in the absence of RPL5. We have also rewritten the figure legend for **Fig. S4d**.

In addition, we have performed another experiment to visualize the sites of active transcription in the nucleolus (by EU incorporation) and UBF, a component of the FC region, and created the new **Fig. S4e**. We pulse-labeled nascent rRNA with 0.5 mM EU for 10 min, and then performed immunofluorescence analyses of UBF in HeLa cells treated with control or RPL5 siRNAs. We found that, in control cells, UBF was localized as large puncta surrounded by nascent rRNAs, indicating that active rDNAs are accumulated and bundled together in the nucleolus. In contrast, UBF and EU-labeled nascent rRNAs were in smaller puncta and dispersed in the RPL5-depleted cells. It is noteworthy that the UBF protein levels are comparable in the control and RPL5 knockdown cells (**Fig. 2b**). Active rDNAs were clustered in the nucleoli of the control HeLa cells, but they were scattered in the RPL5-depleted cells. Maximum diameters of nucleolar UBF staining were also measured, and supported that rDNA compaction was affected in the absence of RPL5. These points are now described in the text, on page 11, line 19-page 20, line 1.

In addition, the authors should address the comments below.

General comment

-Similar published work should be better referenced:

Several medium/high throughput screens aimed at identifying factors important for nucleolar structure maintenance have been performed.

Thank you for pointing out this important issue with your detailed advice. We responded, as below after your comments.

In one work, the eighty ribosomal proteins were systematically investigated for a role in nucleolar structure maintenance, mature rRNA accumulation, pre-rRNA processing, and p53 homeostasis. See DOI:10.1038/ncomms11390 and www.ribosomalproteins.com.

On this occasion an algorithm was specifically developed, the iNo scoring method, which offers unprecedented statistical power, unrivaled thus far for this type of analysis. Half of the conclusions of this submission are in this published work. Conclusions like 'uL18 is the most important ribosomal protein for nucleolar structure maintenance' or 'among proteins of the large ribosomal subunit those that assemble late are the most important ones for nucleolar structure' etc. were already spelled out in this published work, and should be put in perspective here for scholar balance.

We apologize that the previous excellent study was not sufficiently referenced in our manuscript. To provide a perspective between the previous work and our work, we now describe it in the introduction, results and discussion sections, on page 5, lines 3-6, and page 14, lines 4-5, and page 17, lines 6-21. We respectfully agree that the iNo scoring method offers unprecedented statistical power, unrivaled thus far for nucleolar morphological analysis. We are in fact glad that regardless of the fundamentally different experimental set-ups, the previous work and our work here resulted in the common finding that RPL5 is the most important ribosomal protein for nucleolar structure maintenance.

This previous work by Lafontaine's group employed the model-based (iNO scoring method) image analysis system that was specifically developed for nucleolar analyses. In our work, we used the model-free (wndchrn) system, without any step specialized for the nucleolar morphologies, including segmentation or additional algorithm or pipeline construction. Furthermore, we screened proteins among 745 factors that are involved in nuclear events and signaling pathways for gene regulation, energy metabolism and DNA repair (Boulon *et al.*, 2010), together with nucleolar components that had been previously identified by a proteome analysis (Andersen *et al.*, 2005), while this previous work screened eighty ribosomal proteins. Regardless of these differences, RPL5 was commonly found to be an important factor for the nucleolus. These results solidify the significance of RPL5, and suggest the versatility of the wndchrn image system for applications to other structures in the cell. We hope that the reviewer could kindly consider that these points are the novelty of our work and provide important information for researchers.

In a second work the abundant nucleolar proteome (625 proteins) was tested for nucleolar structure; presumably there is a very large overlap between this set and the set of factors tested here. See DOI: 10.1038/s41596-018-0044-3. Again, it would be useful to compare the results (at least briefly).

According to this reviewer's comment, we have compared the results from Dr. Lafontaine's and our groups. We have revised Table S1 by highlighting in yellow the proteins that were tested in both

studies. We also found that five proteins were commonly identified to be important for nucleolar morphologies. Accordingly, we added the following sentences in the text, on page 17, lines 22-page 18, line 3.

“In another study, the same group screened factors that are involved in nucleolar structure maintenance among 668 abundant nucleolar proteins, using the iNo scoring method, and identified 86 proteins as being important (Stamatopoulou *et al.*, 2018). Among them, 52 were also in our siRNA library (highlighted in yellow in **Table S1**). Five of them (RPL5, RPL27A, NPM1, WDR43 and TIF-IA) were among the 16 proteins that we found to be important for nucleolar morphology (**Fig. 1a-b**), again solidifying the importance of RPL5 in the nucleolus. ”

Finally, in the iNo scoring method, a DFC marker (fibrillarlin) was used and it was expressed from an additional locus. However, it was also shown in this work that a GC marker (PES1) detected with antibodies (endogenous protein) -similar to NPM1 used in this work- led to the same conclusions.

We apologize that we did not mention that the work used a GC marker (PES1) detected with antibodies (endogenous protein). We revised the text on page 17, lines 6-10.

Specific comments:

(1) Figure 4: there is an inconsistency with regard the effects on rRNA synthesis. The result in Figure 4 panel b (RTqPCR) shows there is an important reduction of the large precursor detected with the amplicons used (47S). This is in contradiction with a previously published work (<https://www.ribosomalproteins.com/uL18/>) where it was shown that uL18(RPL5) depletion leads to strong accumulation of large precursor accumulation (45S/47S). This is also in contradiction with the authors' own data. Indeed, Figure 4 panel e clearly shows that the 45S is accumulated upon RPL5 depletion. The authors should probe this gel in a northern blotting experiment to detect the 47S and 45S and they will confirm this. The accumulation of 32S, on the other hand, is consistent with former work.

We thank the reviewer for raising this important issue. It helped us to think carefully, and improved our paper. First, according to this comment, we performed a northern blot analysis, and the results are shown in the new **Fig. 4b**. The northern blot revealed that 45S/47S accumulated in the absence of RPL5. We described this on page 14, lines 2-5 in the revised manuscript.

We also performed droplet digital PCR (ddPCR) to measure the 47S pre-rRNA precisely, and show the results in the new **Fig. 4c**. Consistent with the northern blot, the ddPCR result revealed the accumulation of 47S pre-rRNA under the RPL5-depleted conditions. We now describe these

experiments on page 14, lines 5-8. We also added the figure legends for **Fig. 4b and c**, on page 42, lines 2-7, and the Materials and methods section, on page 32, line 8-page 33, line 3. We removed the previous qRT-PCR data, because we could not find an appropriate internal control to calculate the relative Ct values precisely.

(2) Figure 1:

In panel b, TIF1-A and POP4 are rooted together.

In panel a, TIF1-A depletion is very well-known to lead to formation of so-called "nucleolar caps" (these can reasonably be seen in panel a). Since POP4 appears so close on the clustering, should its depletion not also lead to cap formation (not visible on panel a)? May be higher magnification would help addressing this?

We thank the reviewer for this insightful comment. As the reviewer pointed out, the nucleolus in the TIF1A knockdown cells has a cap structure. The nucleolus of the POP4 knockdown also has a cap structure that becomes visible when magnified, as shown on the right. We replaced the old figures for siPOP4 with more appropriate ones in **Fig. 1a**.

[Figure removed by editorial staff per authors' request].

(3) There is a discussion on p53 homeostasis (Fig S3) which is inadequate since this work was performed in HeLa cells (which are not physiologically regulated for p53).

As in our response to the same critique raised by reviewer #2, we performed the same experiment with RPE-1 cells, which were originally derived from normal human retinal pigment epithelial cells and have the normal p53 signaling pathway. The data are now shown in the new **Fig. S3c-e**. In RPE-1 cells, strong p53 protein accumulation was accompanied with p21 accumulation, upon the depletion of TIF-1A due to nucleolar stress (new **Fig. S3c, right**). This indicates that the p53 signaling pathway is functional in this karyotypically normal cell. The p53 protein was not remarkably accumulated upon the RPL5 depletion in RPE-1 cells. Even though, we observed the

aberrant nucleolar morphologies in these cells (new **Fig. S3d**). Cell growth was also inhibited with the RPL5 knockdown in both the HeLa and RPE-1 cell lines (new **Fig. S3e and S3f**). These findings suggest that the cell growth inhibition and aberrant nucleolar morphology upon the RPL5 knockdown are independent of the p53 signaling pathway. These results are now described in the text, on page 8, lines 11-23. We have also created new figure legends for the new **Fig. S3c-e**.

(4) In Figure 3, panel a, DAPI signal: is it just this cell or a more general observation that the DAPI signal is also affected upon uL18 depletion?

We are excited by this comment, because we are quite interested in it. As shown below, dense DAPI signals surround the nucleolar periphery, which corresponds to heterochromatin in control cells (siControl). In the RPL5 knockdown cells (siRPL5), the heterochromatin is absent from the nucleolar periphery. It is a general phenomenon, because most of the 160 pictures show the same result. This may be due to the altered biophysical property of the nucleolus. This is interesting, but it is still a speculation and we decided that it is beyond the scope of the present work.

[Figure removed by editorial staff per authors' request].

(5) Effects on nucleolar structure in DBA cells (Figure 5): I really appreciate the effort to describe the nucleolus in peripheral blood cells, which are more difficult to image. It is quite difficult to see nucleolar morphological alterations on the images. What is clearly apparent is a more intense

labelling of the GC protein NPM1.

> I really appreciate the effort to describe the nucleolus in peripheral blood cells, which are more difficult to image.

We heartily thank the reviewer for their appreciation and understanding of our efforts.

>It is quite difficult to see nucleolar morphological alterations on the images. What is clearly apparent is a more intense labelling of the GC protein NPM1.

The differences in the nucleolar morphologies between the controls and the DBA patient, shown in **Fig. 5c-d**, can be described with the feature values that were useful for the discrimination in the wndchrm analyses. Therefore, we created the new **Table S5**, showing a list of the image features with ranking based on Fisher discrimination scores. These are described in the revised manuscript, on page 15, lines 7-9. Please note that many features were processed for transformations, including Fourier and Chebyshev, suggesting that the difference may not be recognized solely by eye-inspection.

References

- Nicolas E, Parisot P, Pinto-Monteiro C, de Walque R, De Vleeschouwer C, Lafontaine DL (2016) Involvement of human ribosomal proteins in nucleolar structure and p53-dependent nucleolar stress. *Nat Commun* 7: 11390
- Stamatopoulou V, Parisot P, De Vleeschouwer C, Lafontaine DLJ (2018) Use of the iNo score to discriminate normal from altered nucleolar morphology, with applications in basic cell biology and potential in human disease diagnostics. *Nat Protoc* 13: 2387-2406
- Boulon S, Westman BJ, Hutten S, Boisvert FM, Lamond AI (2010) The nucleolus under stress. *Mol Cell* 40: 216-227
- Andersen JS, Lam YW, Leung AK, Ong SE, Lyon CE, Lamond AI, Mann M (2005) Nucleolar proteome dynamics. *Nature* 433: 77-83
- Mitreá DM, Grace CR, Buljan M, Yun MK, Pytel NJ, Satumba J, Nourse A, Park CG, Madan Babu M, White SW *et al.* (2014) Structural polymorphism in the N-terminal oligomerization domain of NPM1. *Proc Natl Acad Sci U S A* 111: 4466-4471
- Mitreá DM, Cika JA, Guy CS, Ban D, Banerjee PR, Stanley CB, Nourse A, Deniz AA, Kriwacki RW (2016) Nucleophosmin integrates within the nucleolus via multi-modal interactions with

- proteins displaying R-rich linear motifs and rRNA. *eLife* 5
- Donati G, Peddigari S, Mercer CA, Thomas G (2013) 5S ribosomal RNA is an essential component of a nascent ribosomal precursor complex that regulates the Hdm2-p53 checkpoint. *Cell Rep* 4: 87-98
- Kressler D, Bange G, Ogawa Y, Stjepanovic G, Bradatsch B, Pratte D, Amlacher S, Strauß D, Yoneda Y, Katahira J *et al.* (2012) Synchronizing nuclear import of ribosomal proteins with ribosome assembly. *Science* 338: 666-671
- Yao RW, Xu G, Wang Y, Shan L, Luan PF, Wang Y, Wu M, Yang LZ, Xing YH, Yang L, Chen LL (2019) Nascent Pre-rRNA Sorting via Phase Separation Drives the Assembly of Dense Fibrillar Components in the Human Nucleolus. *Mol Cell* 76:767-783
- Luo ST, Zhang DM, Qin Q, Lu L, Luo M, Guo FC, Shi HS, Jiang L, Shao B, Li M *et al.* (2017) The Promotion of Erythropoiesis via the Regulation of Reactive Oxygen Species by Lactic Acid. *Sci Rep* 7: 38105
- Kampen KR, Sulima SO, Vereecke S, De Keersmaecker K (2020) Hallmarks of ribosomopathies. *Nucleic Acids Res* 48: 1013-1028
- Gazda HT, Sheen MR, Vlachos A, Choemmel V, O'Donohue MF, Schneider H, Darras N, Hasman C, Sieff CA, Newburger PE *et al.* (2008) Ribosomal protein L5 and L11 mutations are associated with cleft palate and abnormal thumbs in Diamond-Blackfan anemia patients. *Am J Hum Genet* 83: 769-780

February 21, 2022

RE: Life Science Alliance Manuscript #LSA-2021-01045-TR

Dr. Noriko Saitoh
Japanese Foundation For Cancer Research
3-8-31 Ariake, Koto-ku
Tokyo 135-8550
Japan

Dear Dr. Saitoh,

Thank you for submitting your revised manuscript entitled "Ribosomal protein L5 facilitates rDNA-bundled condensate and nucleolar assembly". We would be happy to publish your paper in Life Science Alliance pending final revisions necessary to meet our formatting guidelines.

- please address the final remaining Reviewer 2 and 3 comments
- please upload your main and supplementary figures as single files
- please add ORCID ID for the corresponding (and secondary corresponding) author-you should have received instructions on how to do so
- please add the Twitter handle of your host institute/organization as well as your own or/and one of the authors in our system
- please make sure the author order in your manuscript and our system match
- please be sure that all Authors are listed in the Author Contribution section
- please use Capital letters when introducing panels in the figures, their legends, and callouts in the manuscript text
- please add your main, supplementary figure, and movie legends to the main manuscript text after the references section;
- please add callouts for Figure 6A-B to your main manuscript text
- please add scale bars to figure S4A

A. FINAL FILES:

B. MANUSCRIPT ORGANIZATION AND FORMATTING:

Sincerely,

Reviewer #2 (Comments to the Authors (Required)):

Overall I am satisfied with the revisions. The authors have improved the explanation of the results and logic and citation of the previous related study.

There are a few minor suggestions for improvement.

1. The authors have attempted to address the concerns regarding the suitability of using DBA lymphocytes to understand nucleoli in DBA in general. One way they addressed the issue was by adding data in K562 cells, a leukemic cell model. Both of these cell models seem subpar and the authors could add a sentence acknowledging that these are imperfect models for understanding DBA (page 16).
2. Some of the language would benefit from editing. As an example "eye inspection" could be replaced with "visual inspection."
3. Figure 4F is very dark and the bands are hard to see.

Reviewer #3 (Comments to the Authors (Required)):

We would like to congratulate the authors on their revision.
This has become a nice contribution.

We really appreciated that the published literature was better incorporated so scholar balance was respected.

There would remain one minor change to make: an opposition between a so-called 'model-based' and 'model-free' image analysis is made, which is not very clear, and should be corrected (see In the rebuttal letter in response to reviewer #1, and in the revised manuscript: page 17, lines 19 and 20.)

We agree that the iNo scoring method was purposely designed for scoring nucleolar morphology alterations, while wndchrm was not. But we don't understand what the authors mean exactly when they wrote that Nicolas et al. used a model-based method when they refer to the iNo scoring method? If the authors are implying by 'model-based' that the iNo scoring method is using preconceived classes to produce its output. This is incorrect. The iNo scoring method does not cluster nucleolar alterations phenotypes according to any predetermined classes or categories. In the PCA output, the classes that emerge following iNo

scoring analysis are unsupervised.

We suggest the authors simply remove the notion of 'model-based' and model-free' systems because its really confusing.

2) We appreciated the novel Supplementary Figure 4e (EU labeling). This is a really good addition to the work.

3) Nomenclature: RPL5 versus uL18.

This is really not a major point, and we leave it to the appreciation of the authors to change this or not.

Nonetheless, we would like to mention that more and more colleagues are using the new nomenclature for ribosomal proteins. In this new nomenclature RPL5 is referred to as uL18. This new nomenclature has deep structural grounds and was agreed on by dozens leading investigators in the field a few years ago. The new nomenclature was published in a manuscript (doi: 10.1016/j.sbi.2014.01.002) which has been refereed 458 times so far.

Re: Life Science Alliance manuscript #LSA-2021-01045-TR

Point-by point responses are below. The reviewers' comments are italicized and highlighted in blue. The revised part in the text is highlighted in red.

Reviewer #2 (Comments to the Authors (Required)):

Overall I am satisfied with the revisions. The authors have improved the explanation of the results and logic and citation of the previous related study.

We thank the reviewer for this positive comment.

There are a few minor suggestions for improvement.

1. The authors have attempted to address the concerns regarding the suitability of using DBA lymphocytes to understand nucleoli in DBA in general. One way they addressed the issue was by adding data in K562 cells, a leukemic cell model. Both of these cell models seem subpar and the authors could add a sentence acknowledging that these are imperfect models for understanding DBA (page 16).

We understand the reviewer's viewpoint here. We have added a new sentence in keeping with the reviewer's suggestion, on page 15, line 18, as follows.

“Although the patient lymphocytes and K562 cells are imperfect models, these results indicate that... ”

2. Some of the language would benefit from editing. As an example "eye inspection" could be replaced with "visual inspection."

We have revised the text, accordingly. The manuscript has been English-edited by a native professional.

3. Figure 4F is very dark and the bands are hard to see.

According to this comment, we have replaced it with a brighter gel image in Figure 4F.

Reviewer #3 (Comments to the Authors (Required)):

We would like to congratulate the authors on their revision.

This has become a nice contribution.

We really appreciated that the published literature was better incorporated so scholar balance was respected.

We are greatly encouraged by these positive comments.

There would remain one minor change to make: an opposition between a so-called 'model-based' and 'model-free' image analysis is made, which is not very clear, and should be corrected (see In the rebuttal letter in response to reviewer #1, and in the revised manuscript: page 17, lines 19 and 20.)

We agree that the iNo scoring method was purposely designed for scoring nucleolar morphology alterations, while wndchrn was not. But we don't understand what the authors mean exactly when they wrote that Nicolas et al. used a model-based method when they refer to the iNo scoring method? If the authors are implying by 'model-based' that the iNo scoring method is using preconceived classes to produce its output. This is incorrect. The iNo scoring method does not cluster nucleolar alterations phenotypes according to any predetermined classes or categories. In the PCA output, the classes that emerge following iNo scoring analysis are unsupervised.

We suggest the authors simply remove the notion of 'model-based' and model-free' systems because its really confusing.

We understand the reviewer's viewpoint. We have removed the words, both 'model-based' and model-free' from the main text, on page 17, accordingly.

2) We appreciated the novel Supplementary Figure 4e (EU labeling). This is a really good addition to the work.

We are glad to hear this very positive comment.

3) Nomenclature: RPL5 versus uL18.

This is really not a major point, and we leave it to the appreciation of the authors to change this or not.

Nonetheless, we would like to mention that more and more colleagues are using the new nomenclature for ribosomal proteins. In this new nomenclature RPL5 is referred to as uL18. This new nomenclature has deep structural grounds and was agreed on by dozens leading investigators in the field a few years ago. The new nomenclature was published in a manuscript (doi: 10.1016/j.sbi.2014.01.002) which has been refereed 458 times so far.

We understand the reviewer's viewpoint, and we are willing to contribute to make a new naming system for ribosomal proteins known to the public. We have revised the abstract, on page 2, lines 6-7,

as below. In addition, we have added this new nomenclature, as a new keyword to this manuscript.

“The depletion of RPL5 (**aka. uL18**),

March 2, 2022

RE: Life Science Alliance Manuscript #LSA-2021-01045-TRR

Dr. Noriko Saitoh
Japanese Foundation For Cancer Research
3-8-31 Ariake, Koto-ku
Tokyo 135-8550
Japan

Dear Dr. Saitoh,

Thank you for submitting your Research Article entitled "Ribosomal protein L5 facilitates rDNA-bundled condensate and nucleolar assembly". It is a pleasure to let you know that your manuscript is now accepted for publication in Life Science Alliance. Congratulations on this interesting work.

DISTRIBUTION OF MATERIALS:

Again, congratulations on a very nice paper. I hope you found the review process to be constructive and are pleased with how the manuscript was handled editorially. We look forward to future exciting submissions from your lab.

Sincerely,
